# Multimodal anti-*Helicobacter pylori* effects of *Lactobacillus casei* HY001: Evidence from in vitro and in vivo studies

Zhonghua Lv, Junqing Yu, Xueting Zhang, Lei Zhang, Chunlei Zhang, Xiaoxu Liu, Yu Chang, Hang Yin, Wei Wang, Haidan Zhao, Huicheng Li*

Harbin Pharmaceutical Group Bioengineering Co., LTD., Harbin, China

* lihch@hayao.com

## Abstract

*Helicobacter pylori* (*H. pylori*) is a highly pathogenic microorganism that can cause various gastric diseases. Accumulating evidences have demonstrated probiotics' potential in combating *H. pylori* infections. The aim of this study was to explore the effects and underlying mechanism of *Lactobacillus casei* (*L. casei*) HY001 against the gastric inflammation and gastric microbiota alteration induced by *H. pylori* infection. These results indicated that *L. casei* HY001 significantly inhibited the growth of *H. pylori* SS1, decreased urease activity, and exhibited strong co-aggregation properties with *H. pylori* SS1 in vitro experiments. Furthermore, *L. casei* HY001 was found to be capable of inhibiting the adhesion of *H. pylori* SS1 with AGS cells. Subsequently, Experiments in animals suggested *L. casei* HY001 alleviated gastric inflammation by inhibiting the expression of NF-κ B and reducing pro-inflammatory mediator levels (IL-8, TNF-α, IL-1β, and IL-6). Moreover, the gastric microbiota 16S rRNA gene sequencing analysis revealed that *L. casei* HY001 improved the structure of the gastric microbiota by modulating the abundance of *Firmicutes*, *Bacteroidota* and *Proteobacteria.* Meanwhile, the relative abundances of *Rothia, Clostridium*-sensu-stricto-1, *Alistipe* and *Prevotellacea*-UG-001 were significantly increased. In contrast, the abundances of *Helicobacter, Turicibacter, unclassified – Muribaculaceae, unclassified-Oscillospiraceae* and *Lachnospiraceae* -NK4A136-group were significantly reduced following HY001 intervention. These researches indicated that *L. casei* HY001 improved *H. pylori* SS1 induced gastric mucosal damage in mice, regulated immune factors, enhanced the reduced diversity of gastric *Lachnospiraceae* microbiota caused by *H. pylori* infection, and restored the stability of the microbial community structure.

## 1. Introduction

*Helicobacter pylori*, a highly motile spiral-shaped, Gram-negative bacterium classified within the phylum of *Proteobacteria*, was first discovered and successfully cultured by

**Data availability statement:** All relevant data are within the manuscript and its Supporting information files.

**Funding:** This study was financially supported by the Science and Technology Project of Heilongjiang Province (Grant No. 9232024Y2332). The funder provided support in the form of salaries for authors, but did not have any additional role in the study design, data collection and analysis, decision to publish, or preparation of the manuscript. The specific roles of these authors are articulated in the 'author contributions' section. There was no additional external funding received for this study. The funders had no role in study design, data collection and analysis, decision to publish, or preparation of the manuscript.

**Competing interests:** Huicheng Li is employed by Harbin Pharmaceutical Group Bioengineering Co., LTD. This does not alter our adherence to PLOS ONE policies on sharing data and materials. The authors have declared that no other competing interests exist.

Australian scientists Barry J. Marshall and J. Robin Warren in 1982, a groundbreaking achievement that later earned them the 2005 Nobel Prize in Physiology or Medicine [1,2].It predominantly colonizes the surface of the gastric mucosa and is closely associated with the development of chronic gastritis, peptic ulcers, and cancer. It has been listed as a Class I carcinogen by the World Health Organization (WHO) in 1994 [3]. Currently, conventional treatments for *H. pylori* infection primarily rely on antibiotics, including dual therapy (proton pump inhibitors (PPIs) and amoxicillin), triple therapy (PPIs, clarithromycin, and amoxicillin/metronidazole), and bismuth-based quadruple therapy [4–8]. Although these treatments are effective against *H. pylori* infections, patient compliance is low, and side effects are significant. Furthermore, the extensive use of antibiotics has led to the emergence of new resistant strains [9,10]. Recent studies have found that the combined use of probiotics and antibiotics can mitigate the side effects of antibiotics while enhancing the eradication rate of *H. pylori* [11,12]. However, few single strains effectively inhibit *H. pylori* in clinical practice. Therefore, the discovery of novel probiotics is imperative.

Probiotics are beneficial live microorganisms that colonize the human body and contribute to the microbial composition of specific host sites. Numerous studies have confirmed that probiotics can effectively treat inflammatory gastrointestinal diseases, including acute diarrhea, constipation, and acute enteritis [13,14]. Furthermore, a substantial body of evidence demonstrates that the adjunctive use of probiotics with standard triple antibiotic therapy can effectively treat *H. pylori* infection while mitigating associated side effect [11,15,16]. While the mechanisms of action of probiotics on gastrointestinal regulation have been studied, the alleviative or therapeutic effects of single strains targeting *H. pylori* infections still unknown.

*Lactobacillus casei* (*L. casei*), a key constituent of the intestinal microbiome, serves as a crucial modulator in promoting digestive system homeostasis and maintaining gut health [17,18]. Research has shown that *L. casei* demonstrates remarkable efficacy in reducing blood pressure and cholesterol levels [19], stimulating cellular proliferation [20], fostering antibody production for immune response, augmenting human immunity, preventing carcinogenesis, and impeding tumor growth [21]. Moreover, it exhibits advantageous probiotic effects such as ameliorating lactose intolerance [22] and allergies [23,24]. These probiotic attributes endow *L. casei* with extensive potential applications in the realm of medicine and healthcare. The diverse advantages of *L. casei* have been substantiated, prompting research on its potential to inhibit *H. pylori* infections. For example, several studies have demonstrated the capability of *L. casei* to inhibit *H. pylori* infection. Such as *L. casei* T1 pretreatment alleviated gastric inflammation and prevent gut microbiota alteration caused by *H. pylori* infection [25]. It was demonstrated that viable cells of the *L. casei Shirota* strain exhibited inhibitory activity against *H. pylori* SS1 [26].

In our study, we evaluated six strains of *L. casei* based on their inhibitory effects on *H. pylori* growth, urease activity, and co-aggregation. The strain demonstrating the strongest inhibitory effect on *H. pylori* was identified as *L. casei* HY001. Furthermore, the inhibitory effect of *L. casei* HY001 on the adhesion of *H. pylori* was explored using the AGS cell model. Finally, the potential of *L. casei* HY001 in alleviating

gastric inflammation and regulating gastric microbiota in mice infected with *H. pylori* was studied through pharmacological intervention.

## 2. Materials and methods

### 2.1. Bacterial strains and culture

The strains of *L. casei* 01 (HY001), *L. casei* 02, *L. casei* 03, *L. casei* 04, *L. casei* 05, *L. casei* 06 were isolated from sour milk samples (Samples were collected from Qinghai Province in China.), while were maintained at −80 °C in our laboratory. *L. casei* HY001 was stored at the Chinese Center for Microbial Culture Collection (CGMCC, Beijing, China; CGMCC NO.26837), The strains of *L. casei* were cultured on de Man, Rogosa, and Sharpe (MRS) agar plates at 37 °C for 24 h.

*H. pylori* SS1, obtained from the Guangdong Microorganism Culture Collection Center (Guangzhou, China), was cultivated on improved Columbia blood solid media. Specifically, *H. pylori* SS1 was grown on a mixture of 23.4 g/L (w/v) Columbia blood agar base (Oxoid, UK) and 9.6 g/L (w/v) brain heart infusion (Oxoid, UK), supplemented with 5% (v/v) sheep blood, under microaerophilic conditions (5% $O_2$, 10% $CO_2$, and 85% $N_2$) at 37°C for 48–72 h.

### 2.2. Screening of *Lactobacillus casei* with anti-*H. pylori* activity

**2.2.1. In vitro inhibition test of *Lactobacillus casei* against *H. pylori*.** *H. pylori* inhibition test in vitro for *L. casei* was determined by the agar-well diffusion test [27,28]. *H. pylori* SS1 was used as indicator microorganisms. *H. pylori* SS1 cells were collected, washed twice with PBS, and then the bacterial concentration was adjusted to $10^8$ CFU/mL using BHI to obtain an *H. pylori* SS1 suspensions. *H. pylori* SS1 suspensions were mixed with BHI blood plate medium to achieve a final concentration of $10^7$ CFU/mL, poured into a sterile plate with an Oxford cup, and then the Oxford cups were removed after the medium were coagulated. *L. casei* strains (*L. casei* 01, *L. casei* 02, *L. casei* 03, *L. casei* 04, *L. casei* 05, *L. casei* 06) were injected into the blood plate pores and cultured for 72 h, and then the bacteriostatic zones diameter of *L. casei* strains were measured.

**2.2.2. Urease activity inhibition of *H. pylori* by *Lactobacillus casei* strain.** Urease activity was determined by a modified phenol red method [29]. *H. pylori* SS1 were incubated in improved Columbia blood solid media (BHI), under microaerophilic conditions (5% $O_2$, 10% $CO_2$, 85% $N_2$) at 37 °C for 48–72 h. *H. pylori* SS1 cells were collected, washed twice with PBS, and then the bacterial concentrations were adjusted to $10^7$ CFU/mL using BHI to obtain *H. pylori* SS1 suspensions. The 40 μL *H. pylori* SS1 cells were added to 10 μL supernatants of *L. casei* strains (the ratio of *L. casei* to *H. pylori* was 1:4), while a suspension of *H. pylori* SS1 alone was used as the control group. After 48 h incubation under microaerophilic conditions at 37 °C, the plate was added to 150 μL of urease reaction buffer (20% (w/v) urea and 0.012% phenol red in 0.9% NaCl solution, with the final pH adjusted to 6.8) on a microtiter plate. The plate was incubated at 37 °C for 2 h, and its absorbance was measured at 550 nm with a microplate reader (Thermo, USA). Urease activity of *H. pylori* at 48 h was evaluated by detecting the absorbance of urease test solution at 550 nm.

$$\text{The inhibition rate of Urease activity(\%)} = \frac{(A1 - A0)}{A0} \times 100$$

(1)

which A1 represents the absorbance of the control group and A0 represents the absorbance of the sample group.

**2.2.3. Co-aggregation ability of *L. casei* HY001 for *H. pylori* SS1.** The co-aggregation ability was assessed using the methodology outlined by Juntarachot et al (2023). *L. casei* strains overnight cultures were subjected to centrifugation at 10 000 × g for 1 min and subsequently washed twice with PBS. *L. casei* strains suspensions ($OD_{600nm} = 0.6$) were mixed with *H. pylori* SS1 suspension ($OD_{600nm} = 0.4$), vortex for 10 s, and then incubate at 37°C. The absorption values (A) of the mixed suspension at 600 nm were measured at 2, 5, 21, and 24 h, respectively, as well as the coaggregation ability of the strain with pathogenic bacteria was evaluated.

$$\text{Coaggregation Ability \%} = \frac{\frac{Ax+Ay}{2} - A(x+y)}{\frac{Ax+Ay}{2}} \times 100 \tag{2}$$

Which Ax represents the absorbance of *L. casei* strains, Ay represents the absorbance of *H. pylori* SS1, and A (x + y) represents the absorbance of the mixed suspension.

## 2.3. Determination of adhesion ability of strains to AGS cells

**2.3.1. Cell culture.** AGS cells, an adherent human gastric adenocarcinoma epithelial cell line, were used to further evaluate the antagonistic activity of *L. casei* HY001 against *H. pylori* SS1. AGS cells were obtained from the China Center for Type Culture Collection (CCTCC), cultivated in F-12K medium (Gibco) with 10% heat-inactivated fetal bovine serum (Gibco) and 1% penicillin and streptomycin mixture (Gibco) in a humidified incubator with 5% $CO_2$ at 37°C. The medium was replaced every 2 days until a confluent monolayer was achieved.

**2.3.2. Assays of adhesion by *L. casei* HY001 to AGS cells.** The inhibitory potential of *L. casei* HY001 against the adhesion of *H. pylori* SS1 to AGS cells was investigated, following the methodology described by Pryde et al. (2022). Briefly, three different experimental types were performed:

(i) The competitive assay was conducted by incubating AGS cells ($10^4$ cells) with *L. casei* HY001 ($10^7$CFU/mL) and *H. pylori* SS1 ($10^7$CFU/mL) simultaneously for 2 h.

(ii) The Inhibition assay was conducted by pre-incubating AGS cells ($10^4$ cells) with *L. casei* HY001 ($10^7$ CFU/mL) for 1.5 h, followed by addition of *H. pylori* SS1 ($10^7$ CFU/mL) and further incubation for 2 h.

(iii) The displacement assay was conducted by pre-incubating AGS cells ($10^4$ cells) with *H. pylori* SS1 ($10^7$ CFU/mL) for 2 h, followed by addition of *L. casei* HY001 ($10^7$ CFU/mL) and further incubation for 1.5 h.

Competitiveness was calculated as the percentage of adhesion of the *H. pylori* SS1 added with *L. casei* HY001 relative to the number of *H. pylori* -bound bacteria in the absence of *L. casei* HY001 (control). The adhesion inhibition was expressed as a percentage through the following formula:

$$\text{inhibition of adhesion} = \left(1 - \frac{T_1}{T_2}\right) \times 100 \tag{3}$$

Where $T_1$ and $T_2$ are the percentages of adhesion by *H. pylori* cells in the presence and absence of *L. casei* HY001, respectively. *H. pylori* displacement was expressed as the percentage of adhesion by *H. pylori* SS1 cells in the presence and absence of the *L. casei* HY001, as described above.

**2.3.3. Detection of cytokines in AGS culture.** To determine the possible anti-inflammatory effect of *L. casei* HY001 on AGS cells during *H. pylori* SS1 infection. The cell supernatant obtained in the experiments described in the competitive assay was collected and stored at −20 °C, and the expression levels of interleukin-8 (IL-8) [30], tumor necrosis factor alpha (TNF-α) [31], and interleukin-10 (IL-10) [32] were measured using an enzyme-linked immunosorbent assay (ELISA) kit (Bioswamp, Wuhan, China).

## 2.4. Animal and experimental design

**2.4.1. Probiotic and *Helicobacter pylori* SS1 preparation.** *L. casei* HY001 bacterium suspension preparation: The strain of *L. casei* HY001 was transferred to the liquid MRS medium, and was cultured for 16 hours. The bacterial cells were then harvested by centrifugation at 8,000 × rpm for 3 minutes, washed twice with 0.9% Sodium Chloride Injection,

and the bacterial concentration was adjusted with 0.9% Sodium Chloride Injection to deliver a final dose of $5 \times 10^9$ CFU/mL.

*H. pylori* SS1 bacterium suspension preparation: were incubated in improved Columbia blood solid media (BHI), under microaerophilic conditions (5% $O_2$, 10% $CO_2$, 85% $N_2$) at 37 °C for 48–72 h. *H. pylori* SS1 cells were collected, washed twice with 0.9% Sodium Chloride Injection, and then the bacterial concentrations were adjusted to $8 \times 10^9$ CFU/mL using 0.9% Sodium Chloride Injection to obtain *H. pylori* SS1 suspensions.

**2.4.2. Animal administration.** Seven-week-old female C57BL/6 mice (19–20 g) were obtained from Huazhong Agricultural University Laboratory Animal Center (Wuhan, China), and were housed in a controlled environment (12 h daylight cycle at $22 \pm 2$ °C and $50 \pm 10\%$) with food and water available ad libitum. Animal experimental procedures followed the National Institutes of Health guidelines for the care and use of laboratory animals and were approved by Safety Assessment Center, Hubei Province (Safety Assessment Center (Fu) No. 202310200).

**2.4.3. Experimental design and animal groups.** After 1 week of acclimatization, 30 mice were randomly divided into 3 groups receiving 3 different treatments as described below:

(1) the Control group (NC group), which received only a standard diet and gavaged with 300 μL 0.9% Sodium Chloride Injection throughout the entire testing period; After pre-treatment with 0.9% Sodium Chloride Injection for one week, mice were gavaged with 300 μL 0.9% Sodium Chloride Injection every two days for a total of 7 times.

(2) the model group (HP group), which received only a standard diet and gavaged with 300 μL 0.9% Sodium Chloride Injection throughout the entire testing period; After pre-treatment with 0.9% Sodium Chloride Injection for one week, mice were gavaged with 300 μL *H. pylori* SS1 suspensions every two days for a total of 7 times.

(3) the *L. casei* HY001 intervention group (HY001 group), which received only a standard diet and gavaged with 300 μL *L. casei* HY001 bacterium suspension throughout the entire testing period; After pre-treatment with 0.9% Sodium Chloride Injection for one week, mice were gavaged with 300 μL *H. pylori* SS1 suspensions every two days for a total of 7 times.

Perform fasting procedures the night before intragastric administration of *H. pylori* SS1. After the infection with *H. pylori* SS1 is completed, the animals are kept under normal conditions for another two weeks. Meanwhile, the HY001 group is continuously administered the dry *L. casei* HY001 by gavage, while other groups are given the same amount of normal saline by gavage. Two weeks later, the animal was euthanized by cervical dislocation under carbon dioxide asphyxiation.

The stomachs were dissected and opened along the lesser curvature. After washing in cold sterile normal saline, the stomachs were divided into longitudinal strips and fixed in 10% formaldehyde for hematoxylin eosin staining (H&E) and immunohistochemistry (IHC) or immediately frozen in liquid nitrogen before stored in −80 °C.

## 2.5. Sample collection and detection

**2.5.1. Detection of inflammatory cytokines level.** The blood was collected into microfuge tubes and allowed to coagulate for a duration of 60 min. Subsequently, the samples were centrifuged at a speed of $3000 \times g$ for a period of 10 min, following which the serum was extracted and stored at −80°C. The level of Interleukin-8 (IL-8), Interleukin-1β (IL-1β) [33], Tumor necrosis factor (TNF-α) [34], Interleukin-6 (IL-6) [34], pepsinogen I (PGI) and pepsinogen II (PGII) were measured using an enzyme-linked immunosorbent assay (ELISA) kits (Bioswamp, Wuhan, China).

**2.5.2. Histological and immunohistochemical analyses.** Histological and immunohistochemical analyses were performed following the standard method [14,35]. The gastric tissues were fixed in 10% formaldehyde, followed by paraffin embedding and sectioning at a thickness of 5–7 μm to prepare paraffin sections. Hematoxylin and eosin (H&E) staining was performed according to the standard method. These samples were stained with antibodies specific to anti-NF-κB (servicebio technology (Wuhan) Co., Ltd.) for 24 h at 4 °C, and then probed with a horseradish

peroxidase-labeled goat anti-rabbit secondary antibody, and the reaction was developed with DAB kit (servicebio technology (Wuhan) Co., Ltd.).

### 2.5.3. Gastric microbiota analysis.

The genomic DNA from gastric microbiota of gastric tissue was extracted with the TGuide S96 Magnetic Soil/Stool DNA Kit (Tiangen Biotech (Beijing) Co., Ltd.) in accordance with the manufacturer's protocol. The DNA concentration was quantified by the Qubit dsDNA HS Assay Kit and Qubit 4.0 Fluorometer (Thermo Fisher Scientific, Invitrogen, Oregon, USA). The V3 - V4 hypervariable regions of the bacterial 16S rRNA gene were amplified from genomic DNA extracted from each sample using primers (338F: 5'-ACTCCTACGGGAGGCAGCA-3' and 806R: 5'- GGACTACHVGGGTWTCTAAT-3') as described by Sun et al. (2019). The PCR amplicons were purified using Agencourt AMPure XP Beads (Beckman Coulter, Indianapolis, IN, USA) and quantified with the Qubit dsDNA HS Assay Kit on a Qubit 4.0 Fluorometer (Invitrogen, Thermo Fisher Scientific, Oregon, USA). Following individual quantification, the amplicons were pooled in equal proportions. For the construction of libraries, Illumina NovaSeq 6000 (Illumina, San Diego, CA, USA) was utilized for sequencing. The bioinformatics analysis of this study was conducted using the BMK Cloud (Biomarker Technologies Co., Ltd., Beijing, China) [36,37] Sequences with similarity ≥ 97% were clustered into the same operational taxonomic unit (OTU) by USEARCH for subsequent analysis [36]. The SILVA database (release 132) [38] was utilized to classify and label OTUs based on a naive Bayes classifier implemented in QIIME2 [36], with a confidence threshold of 70%. Alpha diversity indices were computed and visualized using both QIIME2 and R software. Beta diversity metrics were evaluated using QIIME to assess the similarity of microbial community compositions across different samples. Principal Coordinate Analysis (PCoA) based on the binary Jaccard distance matrix was employed to visualize β-diversity. Differences between groups were evaluated using the STAMP software. The raw sequencing reads have been deposited in the NCBI SRA under the accession number PRJNA1358791.

## 2.6. Statistics analysis

The experimental data and retrograde statistical analysis were performed using GraphPad Prism 8.3.0 software and IBM SPSS statistics 27 software. For multivariate analysis of β-diversity, Permutational Multivariate Analysis of Variance (PERMANOVA) was performed on the binary Jaccard distance matrix using the 'adonis2' function (vegan package, v.2.6–4) with 9999 permutations. Pairwise PERMANOVA tests between groups were conducted where the overall model was significant, and p-values were adjusted for multiple comparisons using the Benjamini-Hochberg false discovery rate (FDR) correction. The experimental data were expressed as Mean ± standard deviation. The data fit the normal distribution, One-way ANOVA and Tukey's post hoc test were used to evaluate the significance of differences. $P < 0.05$ indicates significant difference in component data. $P < 0.001$ indicates an extremely significant difference in the component data. Clustering correlation heatmap with signs was performed using the OmicStudio tools at https://www.omicstudio.cn. Statistically significant correlations are denoted as follows: *$p < 0.05$, **$p < 0.01$.

## 3. Results

### 3.1. Anti-*H. pylori* Activity of *Lactobacillus casei In Vitro*

*H. pylori* growth inhibition test is one of the crucial indicators for screening anti-*H. pylori*. In our study, six *L. casei* strains had different inhibitory ability against *H. pylori*, and the diameter of inhibitory zone ranged from 8 mm to 16.17 mm. Among them, *L. casei* 01 exhibited the strongest inhibitory ability against *H. pylori* SS1, with a diameter of 16.17 mm (Table 1).

To further insights into the mechanism by which *L. casei* strains inhibit *H. pylori* SS1, we investigated the effect of *L. casei* strains on urease activity in liquid cultures of *H. pylori* SS1. As shown in Table 1, six strains of *L. casei* have different abilities in inhibiting the urease activity of *H. pylori* SS1, and the rate of urease activity of *H. pylori* SS1 ranged from 21.17 ± 1.00% to 44.78 ± 1.00%. The inhibitory effect of *L. casei* 01 on the urease activity of *H. pylori* SS1 (44.78 ± 1.00%) was found to be significant.

**Table 1. Anti-*H. pylori* activity and urease inhibition of *L. casei*; Data are expressed as mean±SD. There were significant differences between strains represented by different letters (p<0.05).**

| Strain number | Anti-*H.pylori* activity(mm) | Urease inhibition(%) |
|---|---|---|
| *L.casei* 01 | 16.17±0.29a | 44.78±1.00a |
| *L.casei* 02 | 15.00±0.00bc | 37.39±1.73b |
| *L.casei* 03 | 14.50±0.50c | 12.17±1.00d |
| *L.casei* 04 | 15.00±0.50b | 38.04±0.65b |
| *L.casei* 05 | 13.50±0.50d | 29.85±0.39c |
| *L.casei* 06 | 8.00±0.00e | 36.51±1.66b |

Meanwhile, we also investigated the co-aggregation ability of the six strains. The result demonstrated that *L. casei* 01 exhibits the highest co-aggregation ability with *H. pylori*, achieving a rate of 57.22±1.66% within 24 h (Fig 1).

These results revealed *L. casei* 01 as a potential strain with activity against *H. pylori*, designated as *L. casei* HY001. These observations are in agreement with previous studies on *L. casei* T1 [25]

### 3.2. Exclusion of the adhesion of *H. pylori* SS1 by *L. casei* HY001

The exclusion effect of *Lactobacillus* on *H. pylori* SS1 adhesion was explained, including the inhibition, displacement, and competition assays, by constructing an AGS cell model. The result showed that the adhesion ratios of *H. pylori* SS1 were reduced to 87.08±2.19%, 84.64±1.91%, and 88.59±3.77 by *L. casei* HY001, with respect to those of the control group (Fig 2). This indicates that the primary function of *L. casei* HY001 is to significantly reduce the adhesion of *H. pylori* SS1 through displacement.

### 3.3. Anti-inflammatory effect of *L. casei* HY001 against *H. pylori*-infection in AGS cells

To ascertain the anti-inflammatory effect of *L. casei* HY001 on AGS cells, we monitored the levels of cytokines IL-8, TNF-α and IL-10 in the AGS cell culture. This culture was exposed to *H. pylori* SS1 cells while being protected by cells of either *L. casei* HY001. Challenge of AGS cells with *H. pylori* SS1 significantly increased the levels of IL- 8 and TNF-α (Fig 3a and c). The result aligned with the pathogen's ability to induce a potent inflammatory response in gastric cells. The level of IL-8 decreased significantly in *L. casei* HY001 group relative to that in the *H. pylori* SS1 treatment group. By contrast, challenge of AGS cells with *H. pylori* SS1 significantly decreased the levels of IL-10 (Fig 3b, P<0.05). The levels of IL-10

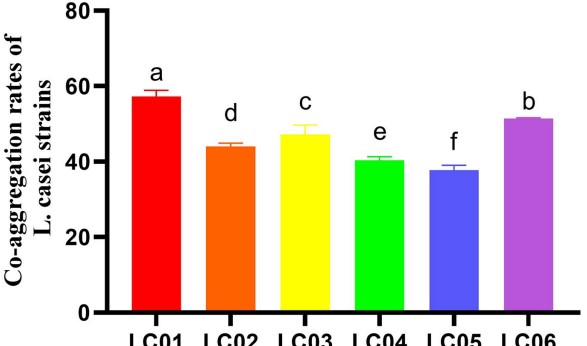

**Fig 1. Co-aggregation rates of *L. casei* strains, with *H. pylori* SS1 in 24 hour.** Data are expressed as mean±SD. There were significant differences between strains represented by different letters (p<0.05).

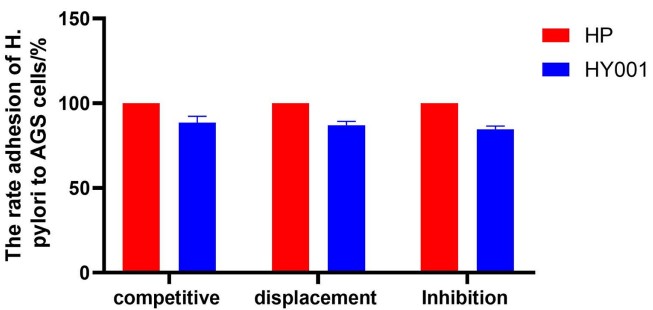

**Fig 2. Anti-adhesion assays (inhibition, displacement and competition) of *L. casei* HY001 against *H. pylori* SS1 on cell line of AGS.** Data are expressed as mean±SD.

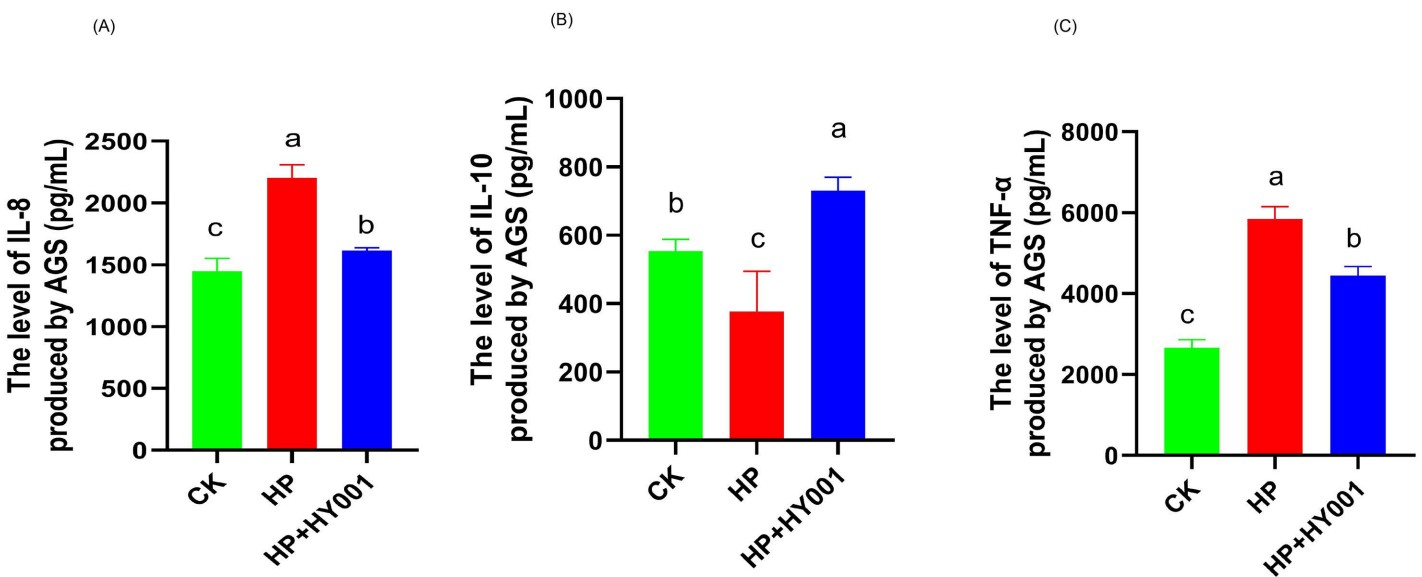

**Fig 3. Determination of IL-8 (a), IL-10 (b), TNF-α (c) levels produced by AGS cells infected with *H. pylori* SS1 and colonized with *L. casei* HY001 and non-infected (control) by using an ELISA assay.** Data are expressed as mean±SD. There were significant differences between strains represented by different letters (p<0.05).

were significantly increased in all the groups of *L. casei* HY001 (*P*<0.05). The supernatant of *L. casei* HY001 remarkably achieved high IL-10 levels during co-incubation in the AGS cell line.

### 3.4. Anti-*H. pylori* SS1 Activity of *L. casei* HY001 *In Vivo*

#### 3.4.1. *L. casei* HY001 pretreatment alleviated gastric mucosal inflammation caused by *H. pylori* SS1 infection in mice.
The effect of *L. casei* HY001 on mice infected with *H. pylori* SS1 was studied to further clarify the mechanism by which *L. casei* HY001 inhibit *H. pylori* SS1. We did not observe any adverse events (such as "severe diarrhea, vomiting, bloating, or infections") related to the intervention of *L. casei* HY001. The mice were euthanized after experimental period over, and a rapid urease test was conducted to determine whether their gastric tissues were infected with *H. pylori* SS1. The bacterium *H. pylori* SS1 can produce a highly potent urease that catalyzes the hydrolysis of urea into ammonia ($NH_3$)

and carbon dioxide ($CO_2$), leading to an elevation in pH, which consequently induces a corresponding red color change in the urease indicator. As shown in Fig 4a, those samples of NC displayed yellow (red indicates positive, yellow indicates negative), whereas those samples of HP were all purple red. The gastric sample of *L. casei* HY001 exhibited a lighter red color compared to the HP group. Quantitative analysis revealed significantly elevated serum urease levels in the HP group (9.548±0.680 ng/mL) compared to normal controls (4.005±0.028 ng/mL) (Fig 4a, $P < 0.001$). The content of urease in serum decreased to (5.610±0.843) ng/mL after HP molding and treatment with *L. casei* HY001, which was close to that of NC group and showed no significant difference ($P > 0.1$). As shown in Fig 4b and 4c, these levels of PGI and PGII were lower in HP group, HY001+HP group than in the control group ($P < 0.01$). These levels of PGI and PGII in serum samples from the HY001 group were significantly elevated compared to those in the HP group ($P < 0.01$).

Additionally, we investigated the effects of *L. casei* HY001 on the mucosal damage resulted from *H. pylori* SS1 infection by conducting histological examination using H&E staining on mouse gastric antrum tissue sections. As shown in Fig 4c, the gastric mucosal glands of the NC group were complete and regular in shape, closely arranged, the structure of gastric mucosa was clear, and there was no obvious inflammatory reaction. After *H. pylori* SS1 infection, the structure of gastric mucosal epithelial cells was loose and abnormal, disordered and irregular in shape. The number of epithelial cells were decreased, and more inflammatory cells were infiltrated in the bottom mucosa.

The expression of NF-κ B was detected by immunohistochemistry (IHC). The expression of NF-κ B positive area in the HP and HY001 group was 7.43-fold and 1.01-fold higher than that in NC group, respectively (Fig 4d and f). The data validate the inhibitory effect of *L. casei* HY001 pretreatment on the expression of NF-κ B induced by *H. pylori* SS1 infection.

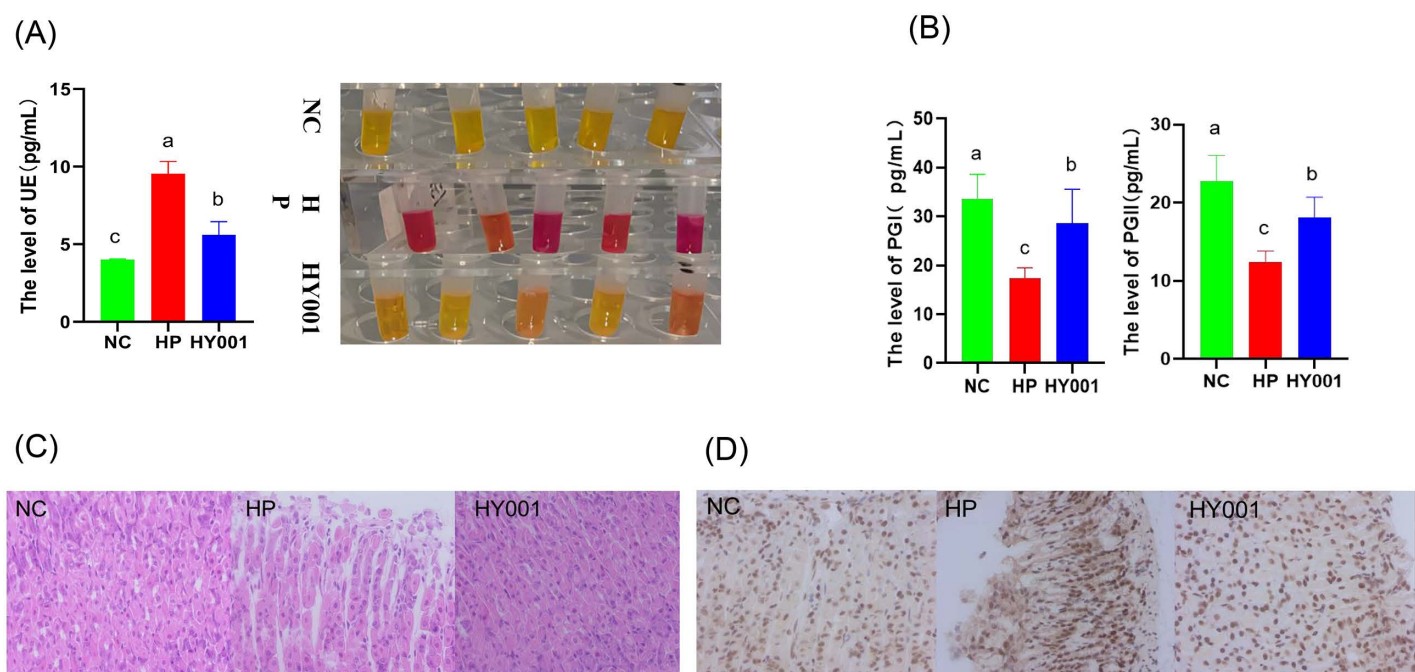

**Fig 4. Effect of *L. casei* HY001 on the expression of gastric mucosal inflammation(n = 5).** (a) the level of UE in serum represents result of the rapid urease test in gastric tissue; (b and c) presents the concentrations of PGI and PGII in serum; (e) shows representative H&E staining of gastric tissue (scale bar: 400 μ m), arrows indicate inflammatory cells infiltration and tissues exfoliation; (d and f) represent expression level of NF-κ B in mouse gastric tissues. Data are expressed as mean±SD. There were significant differences between strains represented by different letters (p < 0.05). HP: treated by *H. pylori* SS1 group vs. pretreated by *L. casei* HY001 and treated by *H. pylori* SS1. NC: untreated group as control; HY001: treatment using *L. casei* HY001 and treatment using H. pylori SS1; HP: treatment using *H. pylori* SS1.

**3.4.2. Effects of *L. casei* HY001 on pro-inflammatory cytokines.** These levels of TNF – α, IL-1β, and IL-6 in the serum of mice were detected after *L. casei* HY001. As shown in Fig 5a-d, *H. pylori* SS1 infection significantly elevated serum IL-8, IL-1β, TNF-α, and IL-6 levels, causing a systemic inflammation state ($P < 0.05$). Treatment with *L. casei* HY001, significantly restored the levels of IL-8, IL-1β, and TNF-α to normal ($P < 0.01$). Although the level of IL-6 was not statistically different in the HY001 group, its average concentration was already lower than that of the HP group. These results indicate that *L. casei* HY001 protected mice from *H. pylori* SS1 induced systemic inflammation.

**3.4.3. Effects of *L. casei* HY001 pretreatment on gastric bacterial diversity and bacterial community alteration.** Sequencing of 16S rRNA amplicon of bacterial microflora in mouse stomach. These changes in gastric microbiota across groups were examined using Principal coordinate analysis (PCoA) based on the binary Jaccard algorithm. The PERMANOVA test, using 9999 permutations on a binary Jaccard distance matrix, indicated that grouping factor explained a significant portion of the variance in β-diversity (pseudo-F = 1.163, R² = 0.162, p = 0.001), with 15 samples distributed across 3 groups. Subsequent pairwise comparisons with FDR correction identified significant differences between NC group and HP group (p-adj = 0.003), and between NC group and HY001 group (p-adj = 0.018), and between HP group and HY001 group (p-adj = 0.012). The greater proximity between the two samples, the higher the degree of similarity in their composition. The proximity between the HY001 group and NC group was reduced, and *H. pylori* SS1 infection exerted a more pronounced impact on the gastric microbial composition of *L. casei* HY001-pretreated mice (Fig 6a).

In addition, to investigate the diversity and abundance of stomach microbial microflora, we conducted Alpha diversity analysis based on OTU feature, as well as analyzed the ACE, Chao1, Shannon and Simpson indexes of each sample. The higher species richness was quantified using the higher ACE and Chao1 indices. The higher species diversity was quantified using the higher Shannon and Simpson indices. As shown in Fig 6 b-e, The ACE index, Chao1 index, Shannon index and Simpson index of HP group all showed significantly decreased ($P < 0.05$). Treatment with *L. casei* HY001, Shannon index and Simpson index already higher than HP group. Although ACE index and Chao1 index were no statistical differences in HP group, The mean concentration in this cohort demonstrated a statistically significant elevation compared to the HP group ($P < 0.05$). *H. pylori* SS1 infection decreased the abundance and diversity. Administration of *L. casei* HY001 improved gastric microbiota structure to near normal levels.

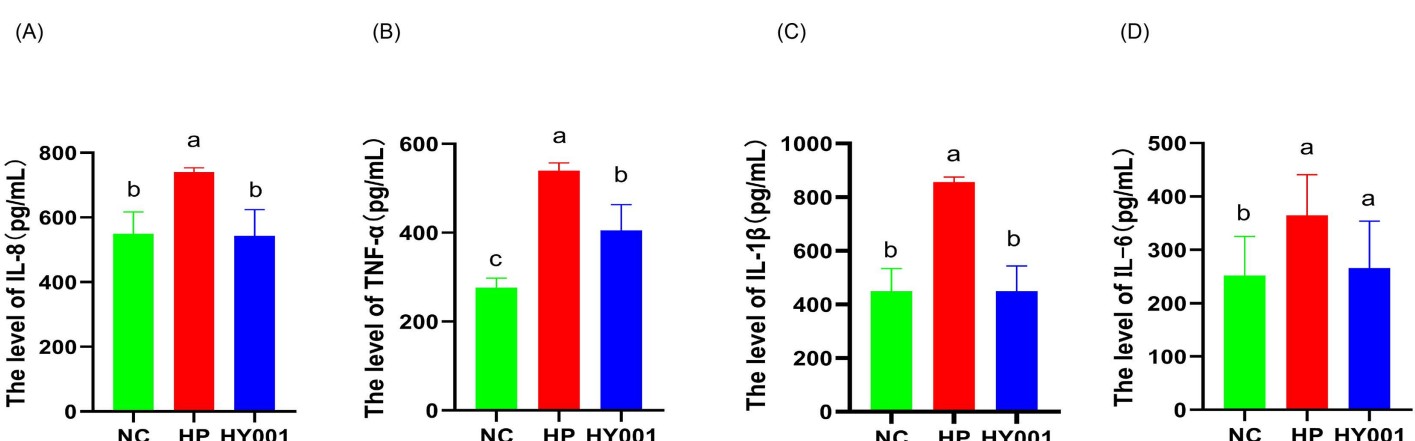

**Fig 5. Determination of IL-8 (a), IL-10 (b), TNF-α (c) and IL-6 (d) level in serum samples of mice infected with *H. pylori* SS1 and colonized with *L. casei* HY001 and non-infected (control) by using an ELISA assay(n = 5).** There were significant differences between strains represented by different letters (p < 0.05).

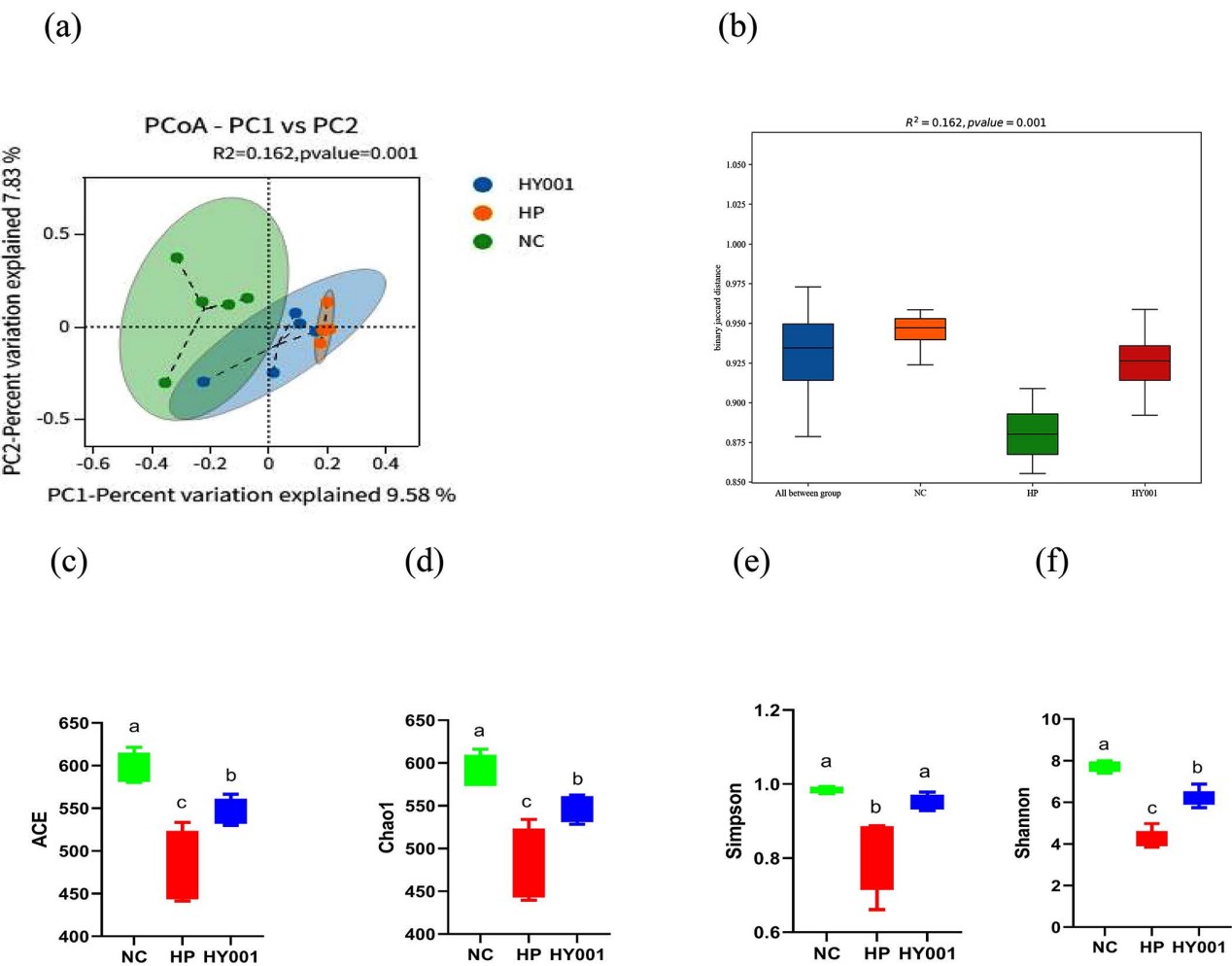

**Fig 6. Effect of _L. casei_ HY001 pretreatment on the microbiota of mouse stomach (n = 5).** (a) Principal coordinate analysis (PCoA) showed that the composition of gastric microorganisms differed between groups. PERMANOVA analyses (b), ACE index (c), Chao1 index (d), Shannon index (e) and Simpson (f). Statistical significance of group separation was assessed by PERMANOVA (pseudo-F = 1.163, R² = 0.162, p = 0.001). Data are expressed as mean ± SD. There were significant differences between strains represented by different letters (p < 0.05).

At the phylum level, the dominant bacterial phyla in the stomach of mice were _Firmicutes_, _Bacteroidota_ and _Proteobacteria_. _Firmicutes_ and _Bacteroidota_ accounted for 75.6% of each sample on average. As shown in (Fig 7a-d), it was found that the gavage of _H. pylori_ SS1 significantly decreased the abundances of _Firmicutes_ an significantly increased the abundance of _Bacteroidota_ and _Proteobacteria_. Intervention with _L. casei_ HY001, the abundances of _Firmicutes_ and decreased _Bacteroidota_ and _Proteobacteria_ increased significantly. In contrast to the NC group, the relative abundance of _Firmicutes_ decreased in the HP group, whereas _Bacteroidota_ was increased.

At the genus level, the relative abundances of bacterial groups (Fig 8a - j), such as _Rothia_, _Clostridium-sensu-stricto-1_, _Alistipe_, unclassified-_cyanobacteriales_, _and Methylocystis_ in the _H. pylori_ SS1 group decreased significantly, while the relative abundance of _Helicobacter pylori_, _unclassified Bacteroidaceae_, _unclassified Oscillospiraceae_, _and Clostridium NK4A136_ increased significantly. Compared with HP group, _L. casei_ HY001 prevention significantly increased the relative abundance of _Rothia_, _Clostridium-sensu-stricto-1_, _Alistipe_, unclassified-_cyanobacteriales_, _and Methylocystis_

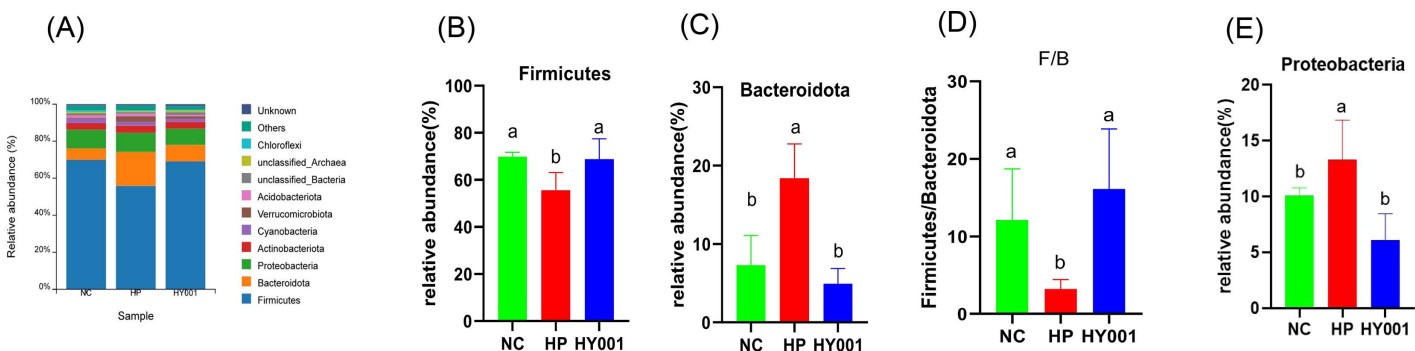

**Fig 7. The effects of *L. casei* HY001 on the microbiota of mouse stomach in mice(n = 5): (a) the phylum level; (b) the relative abundance of *Firmicutes*;(c) the relative abundance of *Bacteroidota*; (d) the relative abundance of *Proteobacteria*.**

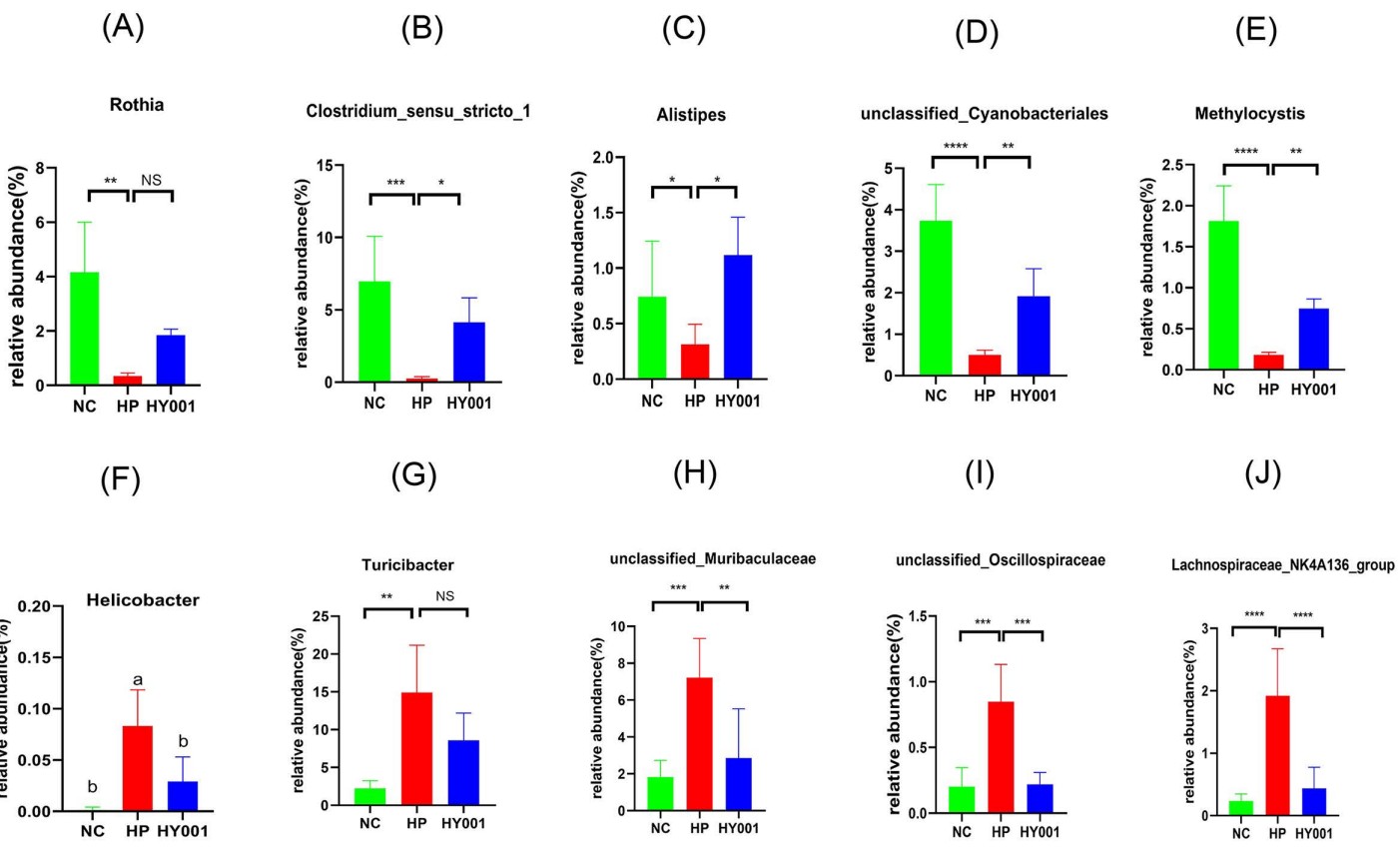

**Fig 8. Relative abundances of top 10 gastric microorganisms at genus level (n = 5); Data are expressed as mean ± SD.** There were significant differences between strains represented by different letters (p < 0.05).

decreased *Helicobacter*, *Turicibacter*, *unclassified-Muribaculaceae*, *unclassified – Oscillospiraceae* and *Lachnospiraceae* -NK4A136-group.

To elucidate the relationships between the gastric microbiota and host inflammatory status, a correlation analysis was conducted between specific bacterial genera and inflammatory factors. The results, as depicted in Fig 9, revealed a

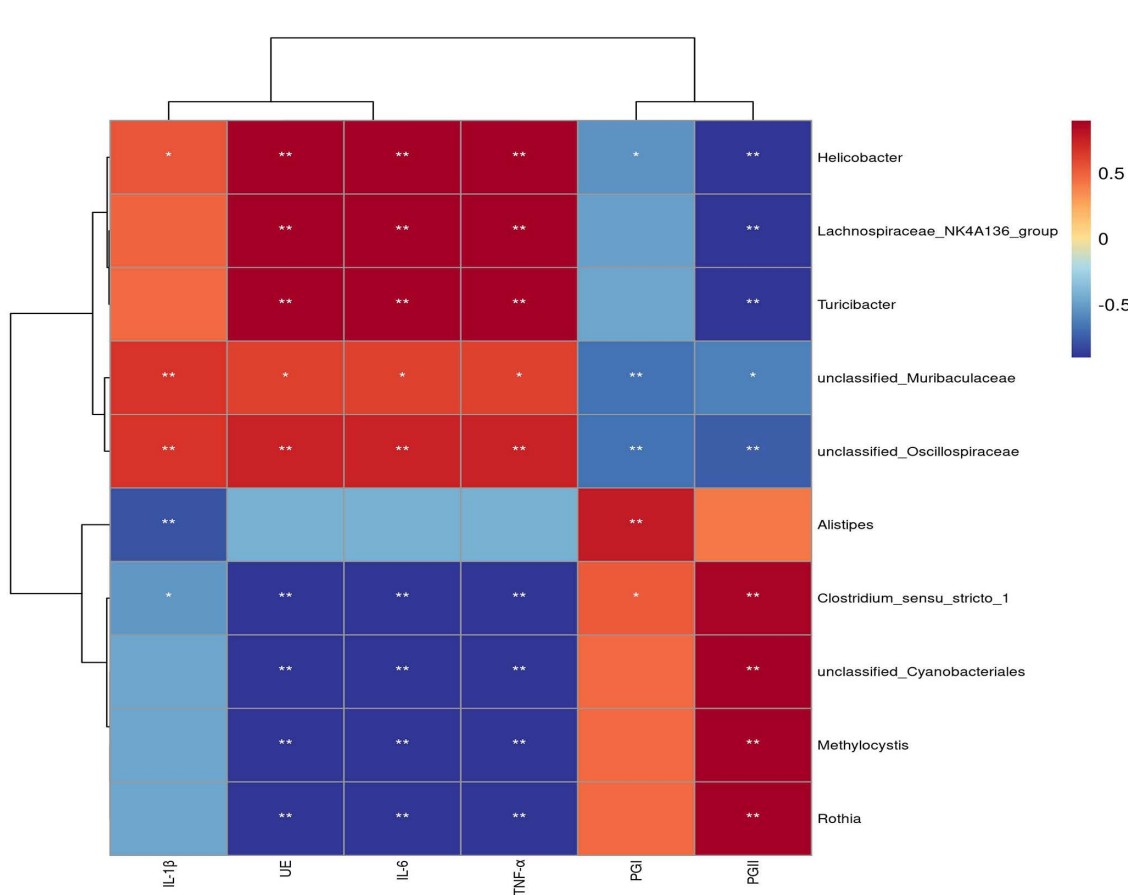

**Fig 9. Correlation heatmap between gastric microbiota and inflammatory factors(n = 5).** The heatmap depicts the Spearman correlation coefficients between selected gastric microbial genera (vertical axis) and inflammatory factors (horizontal axis). Red color indicates a positive correlation, while blue color represents a negative correlation. The intensity of the color corresponds to the strength of the correlation, as shown in the adjacent color bar. Only genera with notable correlation trends are displayed. Statistically significant correlations are denoted as follows: *$p < 0.05$, **$p < 0.01$.

distinct correlation profile. *Helicobacter* demonstrated strong positive correlations with all inflammatory cytokines (IL-1, IL-6, TNF-α). This finding is consistent with the well-established role of *H. pylori* as a primary pathogen that induces host immune responses and chronic gastric inflammation [39]. *Clostridium _ sensu _ stricto _1* and *Turicibacter* also showed positive correlations with the inflammatory factors. Their enrichment suggests a potential role in promoting or sustaining a pro-inflammatory microenvironment within the gastric mucosa. In contrast, *Lachnospiraceae _NK4A136 _group* exhibited significant negative correlations with multiple inflammatory cytokines. As known producers of short-chain fatty acids (SCFAs) with anti-inflammatory properties, this genus is hypothesized to contribute to a protective, homeostatic gastric environment. The genera *Alistipes* and *Rothia* also displayed negative correlation trends with inflammation markers, suggesting they may be part of a beneficial microbial consortium.

## 4. Discussion

*H. pylori* have estimated to infect more than half of the world's population and is a major risk factor for chronic gastritis, peptic ulcer disease, MALT lymphoma and gastric adenocarcinoma [40]. By clarifying the mechanism of *H. pylori*

infection, we can better treat *H. pylori* infection. There are four key steps between the entry of *H. pylori* into the host organism and the onset of disease, as follows: (1) survival in the acidic stomach; (2) movement toward epithelium cells by flagella-mediated motility; (3) attachment to host cells by adhesins/receptors interaction; (4) causing tissue damage by toxin release. Currently, the mainstay of treatment for *H. pylori* infection continues to be antibiotic therapy. Numerous studies have demonstrated that using probiotics can improve the side effects of antibiotics during the treatment of *H. pylori*, thereby increasing eradication rates. In this study, we investigated the effect of *L. casei* HY001 on *H. pylori*-induced gastric inflammation and microbiota. These results demonstrated that HY001 could inhibit the growth of *H. pylori* SS1, alleviate gastric inflammation, and regulate the gastric microecological environment.

*H. pylori* growth inhibition test is one of the crucial indicators for screening anti-*H. pylori* [41]. Studies of probiotics antagonizing *H. pylori* have focused on the inhibitory effects of probiotics on *H. pylori* growth, inhibition of *H. pylori* urease activity and co-aggregation with *H. pylori* [42]. This study used *H. pylori* SS1 with vac A and cag A virulence genes as a model strain, and determined the inhibitory ability of *L. casei* on *H. pylori* SS1 using agar diffusion method. These findings demonstrated that all six strains of *L. casei* exhibited inhibitory effects on the growth of *H. pylori* SS1, with *L. casei* HY001 displaying the most potent inhibitory capacity.

*H. pylori* have a urease on the surface of the bacterium, which can break down urea in the host to produce ammonia, neutralize stomach acid and protect the bacterium from being injured. Some studies have shown that probiotics inhibit the urease activity of *H. pylori* by secreting organic acids. In our study, we found that *L. casei* HY001 could inhibit the urease activity of *H. pylori* SS1 in vitro and in vivo, suggesting that *L. casei* HY001 has potential clinical applications.

*L. casei* HY001 was further evaluated for its ability to co-aggregate with *H. pylori* SS1. Co-agglutination plays an important role in the removal of pathogenic bacteria from the intestine [43]. Probiotics co-aggregate with *H. pylori* can work well to expel *H. pylori* from the body and reduce its load in the body. In this study, *L. casei* HY001 displayed a co-aggregation rate of over 50% and exhibited the capability to agglutinate *H. pylori* SS1 in vitro. *L. paracasei* ZFM54 showed co-aggregation with *H. pylori* (30.28±2.24%) after 24 h of incubation [44].Caterina Holz analyzed that DSM17648 showed co-aggregates the pathogen *in* vivo and in vitro [45].

After the intervention of *L. casei* HY001, a significant reduction in *H. pylori* adhesion to gastric epithelial cells as well as modulation of cytokine expression was observed in AGS cells. The adhesion inhibition of *H. pylori* SS1 by *L. casei* HY001 in C57/BL6 mice was also assessed concurrently. *L. casei* HY001 demonstrated a significant ability to inhibit the adhesion of *H. pylori* in both an AGS cell line model and in the gastric mucosa of C57/BL6 mice, indicating promising preclinical evidence for potential clinical applications.

*H. pylori* attached to host cells via adhesin/receptor interaction, causing tissue damage by toxin release [46]. The first symptom is localized inflammation of the stomach tissue. Probiotics inhibit the growth of *H. pylori* through metabolites, thereby inhibiting the occurrence of stomach inflammation. In addition, the immune pathway of probiotics requires further research.

The NF-κ B pathway plays a pivotal role in the upregulation of pro-inflammatory cytokines during *H. pylori* infection, while also governing the transcriptional network of various inflammatory cytokine genes [47]. Its activation will lead to increased levels of proinflammatory cytokines, such as IL-8 and TNF- α in the gastric mucosa. Studies have shown that a highly virulent strain of *H. pylori*, characterized by its strong Cag A positivity, stimulates gastric epithelial cells to secrete IL-8, a potent chemokine responsible for attracting neutrophils and playing a crucial role in the development of chronic bacterial-induced gastric inflammation [48].

In our studies, those levels of inflammatory factors secreted by AGS cells were significantly elevated following *H. pylori* SS1 infection. While the intervention with *L. casei* HY001 resulted in a decrease in pro-inflammatory factors levels. This result is consistent with that of the mice infected with *H. pylori*. In the treatment of *H. pylori*, the levels of pro-inflammatory factors IL-8, IL-6, TNF-α, and IL-1β were significantly increased in serum. While the intervention with *L. casei* HY001, the levels of pro-inflammatory factors (IL-8, TNF-α, and IL-1β) in these serum samples were all significantly decreased.

Although the decrease in IL-6 did not reach statistical significance, the observed trend is directionally consistent with the reduction in inflammatory factors and our proposed mechanism. We also observed inflammatory changes in *H. pylori*-infected mice, and pathologic sections showed damage to the mucosa of gastric tissue in *H. pylori*-infected mice. In the *L. casei* HY001 intervention group, pathological sections of the gastric tissue mucosa showed less inflammatory damage. PGI is mainly produced by chief cells in the fundic glands [49]. PGI is secreted by the glands in the antrum and cardia of the stomach which secrete mucus [50]. In our studies, in the treatment of *H. pylori*, the levels of PGI and PGII decreased significantly. On the contrary, While the intervention with *L. casei* HY001, the levels of PGI and PGII increased significantly. This result is similar to the report by Yu et al (2023). These results indicate that intervention with *L. casei* HY001 can effectively alleviate gastritis induced by *H. pylori*.

Beyond the immediate impacts observed, probiotics play a key role in maintaining the stability and diversity of the gut microbiota. Maintains the stability and diversity of the gut microbiota, thereby benefiting the host. In this study, at the phylum level, *L. casei* HY001 intervention mice showed a significant decrease in the *Proteobacteria* and an increase in the *Firmicutes*, knotting close to the flora structure of uninfected mice. It was shown that *L. casei* HY001 was able to alleviate the reduction in gastric microflora diversity and restore the flora structure caused by *H. pylori* SS1 infection. *L. casei* HY001 reduces the abundance of harmful flora and increases the abundance of beneficial flora.

*Rothia* is a specialized Gram-positive anaerobic rod-shaped microorganism that exhibits a slight curvature. It possesses the ability to produce short-chain fatty acids (SCFAs) such as acetic acid, propionic acid, and butyric acid, while also demonstrating proficiency in breaking down indigestible carbohydrates [41]. *Rothia* produces a large amount of butyrate from fermentable dietary carbohydrates. *Rothia* primarily parasitizes the mucin layer and ensures butyrate production. As a highly butyrate-producing bacterium, *Rothia* may be important in controlling inflammatory processes, especially in the gastrointestinal tract [51,52]. Clostridium sensu stricto-1 species are capable of fermenting dietary fiber to produce SCFAs, such as butyrate [53,54]. Butyrate serves as a crucial energy source for intestinal epithelial cells, contributing to the maintenance of intestinal barrier function and exerting anti-inflammatory effects [35,55]. *Alistipes* is a producer of propionic acid and acetic acid, exhibiting both pro-inflammatory and anti-inflammatory properties [56]. Meanwhile, the abundance of *Turicibacter* is negatively correlated with acetate [57]. The higher the *Turicibacter* abundance, the lower the acetate content in the gut. Acetate promotes IgA production, inhibits potentially pathogenic bacteria, and protects the intestinal barrier. These results indicate that the *L. casei* HY001 group exhibited a significant increase in the abundance of these strains, suggesting that the HY001 intervention mitigated gastric inflammation caused by *H. pylori* in mice.

Studies showed that the relative abundance of *Helicobacter* in the gastric flora of *H. pylori*-infected mice was significantly elevated. The abundance of *H. pylori* in the group with *L. casei* HY001 intervention decreased significantly and approached that of the NC group. In conclusion, *L. casei* HY001 can increase the abundance of beneficial bacteria and reduce the abundance of harmful bacteria, and regulate the bacterial community in gastric tissues.

Our analysis reveals that the gastric microbiota is structured into distinct, competing ecological modules associated with either inflammation or mucosal health. The strong positive correlations between *Helicobacter, Clostridium _sensu _ stricto _ 1,* and pro-inflammatory cytokines (IL-1β, IL-6, TNF-α) confirm and extend the understanding of a pro-inflammatory microbial consortium beyond *H. pylori* alone. This network likely thrives in and perpetuates a dyspeptic, inflammatory gastric environment [58,59]. Critically, we identified a putative protective module, most notably represented by *Lachnospiraceae _ NK4A136_ group*, which showed significant negative correlations with both inflammation and pepsinogen (PGI / PGII) levels. This genus is a known producer of the anti-inflammatory metabolite butyrate, which can inhibit NF-κB signaling and promote mucosal integrity [60]. The inverse relationship with pepsinogens, established biomarkers of gastric mucosal damage, strongly suggests that this bacterium is a marker of, and potentially a contributor to, gastric homeostasis [61].In conclusion, gastric health appears to be a function of the balance between pro-inflammatory and protective bacterial modules. Therapeutic strategies aimed at suppressing the former while bolstering butyrate-producing genera like *Lachnospiraceae _ NK4A136 _ group* could offer a novel avenue for managing gastritis and its complications.

Based on the aforementioned analysis. The conflicting outcomes in general probiotic therapy against *H. pylori* underscore the critical importance of strain-specific mechanisms. Our findings demonstrate that *L. casei* HY001 possesses a multi-faceted and synergistic mode of action that distinguishes it. Firstly, *L. casei* HY001 employs a direct anti-*H. pylori* strategy by secreting metabolites that inhibit urease activity, a key virulence enzyme, thereby compromising the pathogen's ability to survive in the gastric niche. Secondly, it competes effectively for host cell binding sites, physically displacing the bacterium. Beyond direct antagonism, *L. casei* HY001 modulates the host response by specifically suppressing the NF-κB pathway, thereby alleviating the core inflammatory driver of *H. pylori*-induced gastritis. Finally, and perhaps most profoundly, *L. casei* HY001 exerts a microecological-restoring effect by enhancing overall microbial diversity, suppressing pathobionts, and enriching beneficial taxa. This comprehensive approach—targeting the pathogen's virulence, blocking its adhesion, dampening host inflammation, and restoring a resilient microbial community—highlights the unique and superior potential of strain HY001 as a functional food.

This study has several limitations. First, while no adverse events related to the intervention of *L. casei* HY001 were observed during the study period, and existing literature supports the safety of the probiotic strains employed, our study was limited in its duration and scale for a comprehensive safety assessment. Future large-scale, long-term epidemiological studies are warranted to fully elucidate the safety profile of prolonged probiotic supplementation. Second, the gastric *H. pylori* burden was not directly quantified using colony-forming unit counts or quantitative polymerase chain reaction. Therefore, the conclusions regarding anti–*H. pylori* effects rely primarily on results from the rapid urease test and relative abundance data derived from 16S rRNA gene sequencing, rather than on direct measurements of absolute bacterial load. This methodological constraint should be considered when interpreting the findings. Finally, it should be noted that this study utilized only female mice. While this was necessary to establish a consistent model for the initial validation of HY001's efficacy, it limits the generalizability of our findings. Future studies employing both male and female animals are essential to determine whether the therapeutic effects of HY001 are sex-dependent. Future research incorporating absolute quantification approaches and dual-sex animal models would help further validate the observations reported here.

## 5. Conclusion

In the present study, Experiments in vitro demonstrated that *L. casei* HY001 effectively inhibited the growth of *H. pylori* SS1, suppressed its urease activity, exhibited strong co-aggregation properties with *H. pylori* SS1, and prevented the adhesion of *H. pylori* SS1 to AGS cells. Animal experiments have confirmed that *L. casei* HY001 effectively reduced the indicators related to *H. pylori* infection in the stomach and relieved the inflammatory response in the stomach of mice.

Furthermore, it ameliorated gastric mucosal damage and modulated gastric microflora composition. In conclusion, *L. casei* HY001 effectively reduced the indicators related to *H. pylori* infection in the stomach. Mitigate the inflammatory response induced by *H. pylori* infection modulate the microbial microflora composition in the gastrointestinal tract, improving the infection of *H. pylori*. These provide a theoretical basis for the application of this probiotic in alleviating *H. pylori* infection.

## Supporting information

**S1 Data. Raw data for all experiments and statistical analysis output.** This spreadsheet contains the complete raw data collected during the in vitro and in vivo experiments, including all measurements and experimental conditions and this file contains the detailed results of all statistical tests performed, including ANOVA and post-hoc tests.
(XLSX)

## Author contributions

**Conceptualization:** Zhonghua Lv, Huicheng Li.

**Data curation:** Lei Zhang, Chunlei Zhang, Xiaoxu Liu, Yu Chang, Hang Yin, Wei Wang, Haidan Zhao.

**Formal analysis:** Zhonghua Lv.

**Methodology:** Lei Zhang, Chunlei Zhang, Xiaoxu Liu, Yu Chang, Hang Yin, Wei Wang, Haidan Zhao.

**Project administration:** Huicheng Li.

**Supervision:** Zhonghua Lv, Huicheng Li.

**Validation:** Huicheng Li.

**Writing – original draft:** Xueting Zhang.

**Writing – review & editing:** Junqing Yu, Xueting Zhang, Huicheng Li.

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
