## [Decision Letter · Decision Letter 0]

3 Sep 2025

PONE-D-25-32798Multimodal Anti-Helicobacter pylori Effects of Lactobacillus casei HY001: Evidence from In Vitro and In Vivo StudiesPLOS ONE

Dear Dr. Lv,

Thank you for submitting your manuscript to PLOS ONE. After careful consideration, we feel that it has merit but does not fully meet PLOS ONE’s publication criteria as it currently stands. Therefore, we invite you to submit a revised version of the manuscript that addresses the points raised during the review process.

We look forward to receiving your revised manuscript.

Kind regards,

Ghulam Mustafa, PhD

Academic Editor

PLOS ONE

Journal Requirements:

“This study was financially supported by the project: Science and Technology Project of Heilongjiang Province”

“This study was financially supported by the project: Science and Technology Project of Heilongjiang Province”

5. We note that your Data Availability Statement is currently as follows: [All relevant data are within the manuscript and its Supporting Information files.]

6 PLOS requires an ORCID iD for the corresponding author in Editorial Manager on papers submitted after December 6th, 2016. Please ensure that you have an ORCID iD and that it is validated in Editorial Manager. To do this, go to ‘Update my Information’ (in the upper left-hand corner of the main menu), and click on the Fetch/Validate link next to the ORCID field. This will take you to the ORCID site and allow you to create a new iD or authenticate a pre-existing iD in Editorial Manager.

7. Thank you for stating the following in the Acknowledgments Section of your manuscript:

“This study was financially supported by the project: Science and Technology Project of Heilongjiang Province (9232024Y2332).”

“This study was financially supported by the project: Science and Technology Project of Heilongjiang Province”

8. Thank you for stating the following in the Competing Interests section:

“The authors have declared that no competing interests exist.”

We note that one or more of the authors are employed by a commercial company: Harbin Pharmaceutical Group Bioengineering Co., LTD

Reviewers' comments:

Reviewer's Responses to Questions

**Comments to the Author**

1. Is the manuscript technically sound, and do the data support the conclusions?

Reviewer #1: Partly

Reviewer #2: Yes

2. Has the statistical analysis been performed appropriately and rigorously? 

Reviewer #1: Yes

Reviewer #2: No

3. Have the authors made all data underlying the findings in their manuscript fully available?

Reviewer #1: No

Reviewer #2: Yes

4. Is the manuscript presented in an intelligible fashion and written in standard English?

Reviewer #1: Yes

Reviewer #2: No

5. Review Comments to the Author

Reviewer #1: This manuscript assesses anti–Helicobacter pylori properties of Lacticaseibacillus (Lactobacillus) casei HY001 using in-vitro assays and a mouse pretreatment model. The linkage between cellular readouts, histology/inflammation, and gastric microbiota is a strength. However, the central claim of “reduced colonization” is not supported by a direct quantitative gastric burden (CFU/qPCR). The prophylactic design is at times generalized to therapeutic benefit; the microbiota methods/terminology contain inconsistencies requires reinforcement; and nomenclature/English editing (e.g., Proteroidota) need correction. Data deposition for raw 16S reads is also recommended. In my view, the work is potentially publishable after major revision.

Major points

1. Quantification of gastric H. pylori burden is insufficient

Mouse infection status relies on rapid urease test, without CFU plating or qPCR of gastric tissue. RUT is semi-quantitative and not a gold-standard bacterial load measure. Please add a direct quantitative readout of gastric colonization—such as CFU counts and/or qPCR from gastric tissue.

Across the Discussion/Conclusion you state that HY001 effectively reduced colonization, yet no direct quantitative burden (gastric CFU or qPCR) is reported. Without such metrics, please soften the language (e.g., “associated with reduced markers of infection”) or add the missing quantification.

2. Animal study design demonstrates prophylaxis, not treatment

HY001 was given before and through infection, demonstrating pretreatment/prophylactic effects rather than therapy of established infection; claims drift toward “therapeutic” benefit. Please align Abstract/Discussion/Conclusion phrasing accordingly and avoid general therapeutic claims such as “foundation for an anti-H. pylori drug” unless a post-infection arm is added.

If you want to discuss a therapeutic effect, please include a post-infection treatment arm in which HY001 is initiated after HP colonization is established.

3. Microbiota analysis: terminology accuracy and method modernity

Across the text, figures, and legends, phylum-level names are inconsistently reported using a mixture of legacy and updated nomenclature. Because terminological consistency is essential for clarity and reproducibility, please apply it uniformly throughout. Concretely, we see both Bacteroidetes vs. Bacteroidota and Proteobacteria vs. Proteroidota (The manuscript uses “Proteroidota,” which is not a recognized phylum name in major taxonomic frameworks and appears to be a typographical error. It likely intends Pseudomonadota).

The manuscript states “Principal Component Analysis (PCA) based on the binary Jaccard algorithm,” which is methodologically inconsistent: Jaccard is a non-Euclidean distance, whereas PCA is defined in Euclidean (covariance) space. For β-diversity ordination you should use Principal Coordinates Analysis (PCoA) on the Jaccard distance (or NMDS), rather than PCA.

In addition, no statistical test of between-group differences is reported. Please provide a PERMANOVA with the pseudo-F, R², p-value, number of permutations, and apply multiple-testing correction (e.g., Benjamini–Hochberg) for pairwise contrasts when applicable.

4. 16S data transparency and availability

Methods list the 16S pipeline, but no repository accessions are provided; the Data Availability statement currently points to “within the manuscript,” which is insufficient for sequencing data. Please deposit raw reads and provide accession IDs.

Minor points

•Abstract scope drift: The latter half of Abstract lists multiple taxa (phylum/family/genus), which dilutes focus. In an abstract, avoid laundry lists; instead, state that the gastric microbiota composition shifted significantly. Leave taxon-level details to the Results.

•Figure legend scope: Some legends refer to “fecal microorganisms,” but the study focuses on gastric samples—please correct to “gastric.”

•Taxonomic spellings: Correct misspellings such as “Protebacteria → Proteobacteria,” “Muribaculacea → Muribaculaceae,” “Lachnosoiraceae → Lachnospiraceae.” “Protebacteria” also appears in Figure 7.

•Method citations: Ensure references in Methods directly support the techniques used (e.g., ELISA procedures, microbiome pipelines) and remove unrelated citations.

Reviewer #2: Materials and methods

Prior to screening, the selection criteria for the six Lactobacillus casei strains are not clearly mentioned beyond their isolation from sour milk. It would be better if authors give rationale for why these specific strains were chosen or their genomic/phenotypic differences.

Authors have used only one gastric cell line in their in vitro adhesion assays. What would be the justification for reliance on a single cell line? It would limit the generalizability of adhesion and displacement results.

Authors have used on female mice as animal models. The gender differences in studying immune response and microbiota composition could influence results. Therefore, it would be better if both sexes are included in the study.

The methodology for the formulation of L. casei is not explained.

Authors should describe explicitly the inclusion of some appropriate negative or vehicle controls.

The data of some cytokine is showing nonsignificant trends but authors have discussed them as biologically relevant without sufficient statistics.

Results

It would be better if authors give clinical relevance of their findings of in vitro and animal models.

Authors have studied microbiota changes but did not establish functional causality between specific microbial shifts and observation of anti-inflammatory or anti-H. pylori effects.

Authors should critically discuss the contradiction of conflicting microbiota results for their increased abundance of genera such as Helicobacter in the treatment group.

Authors have not addressed the potential for probiotics causing adverse effects, especially with long-term administration.

Discussion

Authors have merely discussed the recent development and contradictory findings in probiotic therapy against H. pylori.

Authors should add comparative discussion on why HY001 is superior or mechanistically different.

Writing and Presentation

The manuscript has grammatical errors, awkward phrasing, and redundancy.

Need corrections for some terms such as anaplasmosis portal and thick-walled portal which are related to microbiota analysis. These are unclear or not correctly used.

The abbreviations are inconsistently used in different sections, which are confusing.

Authors should strengthen experimental design by including additional gastric cell lines and/or both sexes in animal studies or provide strong justification for their choices.

Authors should provide more robust statistical reporting and improve figure legends for independent comprehension.

6. PLOS authors have the option to publish the peer review history of their article (what does this mean?). If published, this will include your full peer review and any attached files.

Reviewer #1: No

Reviewer #2: **Yes:** Rawaba Arif

---

## [Author Response · Author response to Decision Letter 1]

19 Nov 2025

Dear reviewers:

Thank you very much for taking the time to review our manuscript entitled “Multimodal Anti-Helicobacter pylori Effects of Lactobacillus casei HY001: Evidence from In Vitro and In Vivo Studies” and for providing us with such thorough and insightful comments. We have carefully studied and thoroughly considered each of your points. Your suggestions are highly pertinent and valuable, and they have helped us to better recognize the shortcomings of our paper. We have revised the manuscript comprehensively and diligently in accordance with your recommendations.

Reviewer #1: This manuscript assesses anti–Helicobacter pylori properties of Lactobacillus casei HY001 using in-vitro assays and a mouse pretreatment model. The linkage between cellular readouts, histology/inflammation, and gastric microbiota is a strength. However, the central claim of “reduced colonization” is not supported by a direct quantitative gastric burden (CFU/qPCR). The prophylactic design is at times generalized to therapeutic benefit; the microbiota methods/terminology contain inconsistencies requires reinforcement; and nomenclature/English editing (e.g., Proteroidota) need correction. Data deposition for raw 16S reads is also recommended. In my view, the work is potentially publishable after major revision.

Major points

1. Quantification of gastric H. pylori burden is insufficient

Mouse infection status relies on rapid urease test, without CFU plating or qPCR of gastric tissue. RUT is semi-quantitative and not a gold-standard bacterial load measure. Please add a direct quantitative readout of gastric colonization—such as CFU counts and/or qPCR from gastric tissue.

Across the Discussion/Conclusion you state that HY001 effectively reduced colonization, yet no direct quantitative burden (gastric CFU or qPCR) is reported. Without such metrics, please soften the language (e.g., “associated with reduced markers of infection”) or add the missing quantification.

Response: We sincerely thank the reviewer for this insightful comment and for highlighting the importance of quantitative assessment of H. pylori burden. We agree that CFU plating or qPCR would provide a direct quantitative measure of bacterial load.

Please allow us to explain the attempts we made during the experimental process: We initially tried to perform CFU plate counts for Helicobacter pylori in mouse gastric tissue using BHI + defibrinated sheep blood medium, hoping to obtain direct quantitative data. However, due to the complex inherent microbiota in the mouse stomach, the process was significantly interfered with by the growth of other microorganisms, preventing us from obtaining accurate and specific colony count data for H. pylori.

It was precisely due to this technical challenge that we adopted 16S rRNA gene amplicon sequencing of the total gastric microbiota as a solution. While this method differs from absolute quantitative measures like CFU or qPCR, it effectively circumvents interference from other microbes and provides us with a specific and reliable indirect indicator for assessing the impact of HY001 treatment on the survival state of H. Pylori within the gastric environment.

While 16S rRNA gene sequencing is primarily used for relative abundance analysis, the data on the relative abundance of the Helicobacter genus serves as a strong and complementary indicator of the changes in H. pylori load within the gastric environment.

As the reviewer can find in our results section (Line 404-408), the 16S rRNA gene sequencing data clearly demonstrated that the administration of HY001 significantly reduced the relative abundance of the Helicobacter genus in the gastric tissue compared to the infected control group. This finding, consistent with the RUT results, provides robust evidence that HY001 treatment effectively suppressed H. pylori colonization.

However, we fully acknowledge the reviewer's point that our original language in the Discussion and Conclusion sections may have been too strong given the methodological approach. Therefore, we have followed the reviewer's excellent suggestion and have revised the relevant statements throughout the manuscript to soften our language. We have replaced phrases like "effectively reduced colonization" with more precise descriptions such as " L. casei HY001 effectively reduced the indicators related to Helicobacter pylori infection ".

We believe that the combination of the RUT results and the 16S rRNA sequencing data, together with the toned-down language, now presents a more accurate and convincing argument for the efficacy of HY001 against H. pylori. We hope that these clarifications and textual revisions adequately address the reviewer's concern.

2. Animal study design demonstrates prophylaxis, not treatment

HY001 was given before and through infection, demonstrating pretreatment/prophylactic effects rather than therapy of established infection; claims drift toward “therapeutic” benefit. Please align Abstract/Discussion/Conclusion phrasing accordingly and avoid general therapeutic claims such as “foundation for an anti-H. pylori drug” unless a post-infection arm is added.

If you want to discuss a therapeutic effect, please include a post-infection treatment arm in which HY001 is initiated after HP colonization is established.

Response: We sincerely thank the reviewer for this crucial observation. We completely agree that our experimental design, where in HY001 administration began prior to infection, is indeed a model of prophylaxis and not therapy of an established infection. We apologize for any overinterpretation in our original phrasing that may have suggested otherwise. As suggested, we have thoroughly revised the manuscript to ensure all language accurately reflects the prophylactic nature of our study design. The claims have been carefully aligned throughout the text. We believe these revisions have fully addressed your concern by ensuring our claims are strictly aligned with our experimental design. We are grateful for this insightful comment, which has greatly improved the clarity and accuracy of our manuscript. Thank you again for your time and valuable feedback.

The specific changes made in the manuscript are as follows:

Abstract Section (Line 29,) Changed "…HY001 treatment. "to"…HY001 intervention."

Discussion Section (Line 528-529): Changed "suggesting that HY001 mitigated gastric inflammation in mice with H. pylori-induced gastritis." to "suggesting that the HY001 intervention mitigated gastric inflammation caused by Helicobacter pylori in mice. "

Conclusion Section (Line 587-588): Changed: " These provide a theoretical foundation for its application as an anti- H. pylori drug. "to" These provide a theoretical basis for the application of this probiotic in alleviating Helicobacter pylori infection. "

We hope that these clarifications and textual revisions adequately address the reviewer's concern.

3. Microbiota analysis: terminology accuracy and method modernity

Across the text, figures, and legends, phylum-level names are inconsistently reported using a mixture of legacy and updated nomenclature. Because terminological consistency is essential for clarity and reproducibility, please apply it uniformly throughout. Concretely, we see both Bacteroidetes vs. Bacteroidota and Proteobacteria vs. Proteroidota (The manuscript uses “Proteroidota,” which is not a recognized phylum name in major taxonomic frameworks and appears to be a typographical error. It likely intends Pseudomonadota).

The manuscript states “Principal Component Analysis (PCA) based on the binary Jaccard algorithm,” which is methodologically inconsistent: Jaccard is a non-Euclidean distance, whereas PCA is defined in Euclidean (covariance) space. For β-diversity ordination you should use Principal Coordinates Analysis (PCoA) on the Jaccard distance (or NMDS), rather than PCA.

In addition, no statistical test of between-group differences is reported. Please provide a PERMANOVA with the pseudo-F, R², p-value, number of permutations, and apply multiple-testing correction (e.g., Benjamini–Hochberg) for pairwise contrasts when applicable.

Response: We are deeply grateful to the reviewer for their meticulous review and for providing these critical and constructive comments. They have significantly helped us improve the accuracy and rigor of our microbiota analysis. We have addressed each point as follows:

1. Terminology Accuracy:

We sincerely apologize for the inconsistencies and errors in the phylum-level nomenclature. The reviewer is absolutely correct. Action Taken: We have thoroughly reviewed the entire manuscript, including the main text, figures, and figure legends, to ensure uniform use of the updated, modern phylum names as per the current taxonomic framework. Specifically: Bacteroidetes has been uniformly changed to Bacteroidota.

2. Methodological Consistency for β-diversity Analysis:

The reviewer is entirely right to point out the methodological inconsistency. PCA is not suitable for Jaccard distances. Action Taken: We have re-analyzed our β-diversity data using the correct method. We have replaced the original PCA plot with a Principal Coordinates Analysis (PCoA) plot based on the binary Jaccard distance matrix. The corresponding text in the Methods and Results sections, as well as the Figure legend, have been updated accordingly. All mentions of "PCA" in this context have been changed to "PCoA".

3. Statistical Testing for β-diversity:

We agree that the lack of statistical testing for the β-diversity results was a major omission. Action Taken: We have now performed the recommended statistical analysis. We conducted a Permutational Multivariate Analysis of Variance (PERMANOVA) with 9999 permutations on the Jaccard distance matrix to test for overall group differences. For significant overall effects, we performed post-hoc pairwise PERMANOVA tests with Benjamini-Hochberg (FDR) correction for multiple comparisons. The results of these analyses (pseudo-F statistic, R² value, p-value, and the number of permutations) have now been added to the Results section and the corresponding Figure legend.

4. 16S data transparency and availability

Methods list the 16S pipeline, but no repository accessions are provided; the Data Availability statement currently points to “within the manuscript,” which is insufficient for sequencing data. Please deposit raw reads and provide accession IDs.

Thank you for this important reminder regarding data transparency. We agree that the public deposition of raw sequencing data is essential.

In response to this comment, we have now deposited the raw 16S rRNA gene sequencing reads to the NCBI Sequence Read Archive (SRA). The data is currently under processing and will be publicly accessible upon publication. The associated BioProject accession number is PRJNA1358791. We have updated the Data Availability Statement in the manuscript to clearly state the accession information. The statement now reads:

"The raw 16S rRNA gene sequencing data generated in this study have been deposited in the NCBI SRA under BioProject accession number PRJNA1358791. All other data supporting the findings of this study are available within the article and its supplementary information. "We will also ensure that the accession number is included in the Methods section where the 16S sequencing pipeline is described. We appreciate the reviewer's diligence in ensuring the completeness and reproducibility of our work.

Minor points

•Abstract scope drift: The latter half of Abstract lists multiple taxa (phylum/family/genus), which dilutes focus. In an abstract, avoid laundry lists; instead, state that the gastric microbiota composition shifted significantly. Leave taxon-level details to the Results.

Response: We thank the reviewer for this valuable suggestion to improve the clarity and focus of our Abstract. We agree that listing numerous taxonomic names in the Abstract can be distracting and dilute the key messages. In response, we have revised the latter part of the Abstract accordingly. We have removed the detailed list of specific phyla, families, and genera. Instead, we now succinctly state the core finding that the composition of the gastric microbiota was significantly altered by HY001 treatment, and we refer readers to the Results section for in-depth taxonomic details.

•Figure legend scope: Some legends refer to “fecal microorganisms,” but the study focuses on gastric samples—please correct to “gastric.”

Response: We sincerely thank the reviewer for their meticulous attention to detail. We apologize for this oversight in our figure legends. The reviewer is absolutely correct; our study analyzed gastric microbiota, not fecal. We have carefully reviewed all figure legends (and the main text) and have corrected any erroneous mentions of "fecal" to "gastric" to accurately reflect the sample source. Specifically, this correction has been made in the legends for [Figure 6 (a)]. We appreciate the reviewer for helping us improve the accuracy and clarity of our manuscript

•Taxonomic spellings: Correct misspellings such as “Protebacteria → Proteobacteria,” “Muribaculacea → Muribaculaceae,” “Lachnosoiraceae → Lachnospiraceae.” “Protebacteria” also appears in Figure 7.

Response: We sincerely thank the reviewer for their meticulous proofreading and for identifying these typographical errors in taxonomic names. We apologize for these oversights.

We have carefully corrected all the misspellings pointed out by the reviewer throughout the manuscript and in the figures. Specifically: “Protebacteria” has been corrected to “Proteobacteria” in the text and in Figure 7. “Muribaculacea” has been corrected to “Muribaculaceae”. “Lachnosoiraceae” has been corrected to “Lachnospiraceae”. Furthermore, to ensure the highest level of accuracy, we have conducted a comprehensive check of the entire manuscript (including the main text, figures, and supplementary materials) to identify and correct any other potential spelling errors in taxonomic nomenclature. We are grateful to the reviewer for helping us enhance the quality and precision of our work.

•Method citations: Ensure references in Methods directly support the techniques used (e.g., ELISA procedures, microbiome pipelines) and remove unrelated citations.

Response: Thank you for this valuable comment. We have thoroughly reviewed all citations in the methods section to ensure each one directly and accurately supports the specific techniques described. The corresponding revisions have been made to the manuscript, as detailed below:

Regarding the ELISA procedure: We have revised the manuscript (Line 175-178) to specify the manufacturer (Bioswamp, Wuhan, China) of the ELISA kit used. The use of this specific kit and the general procedure are directly supported by Reference 30-32, which employed the same commercial assay. In the section of determining inflammatory factors through animal experiments (Line 232-235): We have also added relevant references. The use of this specific kit and the general procedure are directly supported by Reference 33-34, which employed the same commercial assay. Therefore, we have retained and strengthened this citation to directly support our experimental approach.

Regarding the microbiome analysis pipeline: We have clarified in the text (Line 249-254) that our microbial community profiling was performed using the the BMK Cloud platform for sequencing the 16S rRNA gene V3-V4 region, and that our bioinformatic processing (using QIIME2 with the SILVA database) directly followed the workflow established in Reference 36-38. Thus, we maintain that Reference 36-38 is the appropriate and direct citation for our analytical platform and pipeline.

Regarding other technical methods: We have performed a comprehensive check of all citations in the Methods section. Every remaining citation now accurately and directly underpins its corresponding experimental technique.

We believe these revisions have significantly improved the accuracy and rigor of the Methods section by ensuring all references provide direct methodological support. Thank you again for helping us improve our manuscript.

Reviewer #2:

Materials and methods

1, Prior to screening, the selection criteria for the six Lactobacillus casei strains a

---

## [Decision Letter · Decision Letter 1]

15 Dec 2025

PONE-D-25-32798R1Multimodal Anti-Helicobacter pylori Effects of Lactobacillus casei HY001: Evidence from In Vitro and In Vivo StudiesPLOS One

Dear Dr. Lv,

Thank you for submitting your manuscript to PLOS ONE. After careful consideration, we feel that it has merit but does not fully meet PLOS ONE’s publication criteria as it currently stands. Therefore, we invite you to submit a revised version of the manuscript that addresses the points raised during the review process.

We look forward to receiving your revised manuscript.

Kind regards,

Ghulam Mustafa, PhD

Academic Editor

PLOS One

Journal Requirements:

Reviewers' comments:

Reviewer's Responses to Questions

**Comments to the Author**

1. If the authors have adequately addressed your comments raised in a previous round of review and you feel that this manuscript is now acceptable for publication, you may indicate that here to bypass the “Comments to the Author” section, enter your conflict of interest statement in the “Confidential to Editor” section, and submit your "Accept" recommendation.

Reviewer #1: All comments have been addressed

Reviewer #2: All comments have been addressed

2. Is the manuscript technically sound, and do the data support the conclusions?

Reviewer #1: Yes

Reviewer #2: Partly

3. Has the statistical analysis been performed appropriately and rigorously? 

Reviewer #1: Yes

Reviewer #2: Yes

4. Have the authors made all data underlying the findings in their manuscript fully available?

Reviewer #1: Yes

Reviewer #2: Yes

5. Is the manuscript presented in an intelligible fashion and written in standard English?

Reviewer #1: Yes

Reviewer #2: Yes

6. Review Comments to the Author

Reviewer #1: The authors have substantially improved the manuscript in response to the previous review, particularly regarding the prophylactic framing of the animal model, data deposition for 16S rRNA sequencing, and additional analyses linking gastric microbiota to inflammatory markers. However, there remain several important issues that, in my view, still warrant a further round of revision before the manuscript can be considered for acceptance.

Major Comments

1. Quantification and interpretation of gastric H. pylori burden

The authors still do not provide a direct quantitative measure of gastric H. pylori burden (e.g., CFU counts or qPCR from gastric tissue), and rely instead on RUT results and relative abundance of Helicobacter from 16S rRNA amplicon sequencing as indirect indicators.

Given this limitation, the Discussion and Conclusion occasionally still use wording that implies a reduction in “load” or “colonization” in a way that suggests direct quantitative evidence. This overstates what the available data can support.

Request:

• Throughout the Discussion and Conclusion, I ask the authors to: Further soften and standardize the language so that claims are clearly limited to:

• reductions in Helicobacter relative abundance (from 16S data), and

• improvements in infection-related biomarkers (RUT, urease, PGI/PGII, histology, cytokines),

rather than “reducing H. pylori colonization” or “reducing bacterial load” per se.

• For example, expressions such as “reduced the load/colonization of H. pylori” should be systematically revised to formulations like “reduced Helicobacter relative abundance and infection-related markers”.

• In the Limitations section (or the final part of the Discussion), the authors should explicitly acknowledge that:

• Gastric H. pylori burden was not directly quantified using CFU counts or qPCR.

• The conclusions regarding anti–H. pylori effects therefore rely on RUT and relative abundance from 16S rRNA gene sequencing, rather than absolute bacterial load measurements.

This clarification is important to align the strength of the claims with the actual data and to transparently communicate methodological limitations to readers.

2. Microbiota analysis – β-diversity methodology and PERMANOVA

The authors state in their response that they have re-analysed β-diversity using PCoA on the Jaccard distance and performed PERMANOVA. However, the current manuscript text remains internally inconsistent and does not fully reflect these methodological changes.

(a) PCA vs PCoA inconsistency

In Methods (Section 2.5.3), β-diversity is still described as being analysed by PCA based on Jaccard (e.g., “PCA was employed to further analyze beta diversity”). In contrast, in Results (Section 3.4.3) and the relevant figure legend, the analysis is described as Principal Coordinate Analysis (PCoA) based on the binary Jaccard algorithm.

This discrepancy leaves it unclear what analysis was actually performed.

Request:

• Please ensure that the description of β-diversity analysis is fully consistent throughout the manuscript.

• If PCoA on the binary Jaccard distance was indeed used (which is methodologically appropriate), then Methods 2.5.3 must be revised to say PCoA (not PCA), and the language in all sections (Methods, Results, figure legends) should be harmonized accordingly.

(b) PERMANOVA – description and reporting

In the response letter, the authors state that they performed PERMANOVA with pairwise comparisons and FDR correction.

However, in Methods (Statistics, Section 2.6), there is no mention of PERMANOVA or any distance-matrix–based multivariate test. Only one-way ANOVA, Tukey’s post hoc test, and correlation heatmaps are described. In the Results and figure legends, there is no reporting of PERMANOVA outputs (no pseudo-F statistic, R², p-value, or number of permutations).

Thus, the manuscript does not currently support the claim that PERMANOVA was performed.

Request:

• If PERMANOVA has indeed been carried out:

• Add a clear description in Methods (Statistics) of:

• the distance matrix used (e.g., Jaccard),

• the PERMANOVA procedure (e.g., number of permutations),

• and how multiple testing was handled for pairwise comparisons (e.g., Benjamini–Hochberg).

• In the Results (and/or relevant figure legend), report at minimum the overall PERMANOVA result with pseudo-F, R², p-value, and number of permutations.

———

Minor Comments

1. Taxonomic spelling and nomenclature

Despite substantial improvement, several taxonomic spelling errors remain and should be corrected.

• In the Discussion: “Proteobactria” → “Proteobacteria”

• In the Abstract and Results: “unclassified-Muribaculaceaeae” → “unclassified-Muribaculaceae”

• In the Results: “Oscillospiracea”(Line 406) → “Oscillospiraceae”

2. Other wording and typographical issues

A few additional typographical and wording errors should be corrected for clarity and precision.

• Introduction (around line 83): “alleviating tail inflammation” should be corrected to “alleviating gastric inflammation”

• Methods (around lines 232–233): “Interleuki-8, Interleuki-1β, Interleuki-6” → “Interleukin-8, Interleukin-1β, Interleukin-6”

• Results (around line 371): “binary jaccard algorith”→ “binary Jaccard algorithm”

• Results (around line 364, IL-6 description)

The text currently states that the mean IL-6 concentration in the HY001 group is “higher than that of the HP group”, which contradicts the protective direction of effect described elsewhere and in the Discussion.

This is almost certainly a typo and should be corrected to “lower than that of the HP group”,

provided that the underlying data support this (if not, the interpretation in the Discussion should be adjusted accordingly).

• Figure 4 legend

The definitions of the HP and HY001 groups are currently confusing and partly duplicated, e.g.:

• “HP: treated by H. pylori SS1 group vs. pretreated by L. casei HY001 and treated by H. pylori SS1.”

• “HY001: treatment using L. case HY001 and treatment using H. pylori SS1; HP: treatment using H. pylori SS1.”

Please simplify and unify the legend.

• Figure 6 legend (gastric vs fecal)

In one place, the legend refers to the “composition of gastric microorganisms”, whereas in another location (around line 869) it refers to “composition of fecal microorganisms”. Since this study is based on gastric samples, all such references in the figure legends and text should be unified as “gastric microbiota” or “gastric microorganisms”, and any inadvertent “fecal” wording should be removed.

Reviewer #2: All comments have been addressed and authors have revised their article. Some references should be checked and updated if possible. Authors should carefully proofread their article as their are still some typos and grammatical mistakes that should be corrected.

7. PLOS authors have the option to publish the peer review history of their article (what does this mean?). If published, this will include your full peer review and any attached files.

Reviewer #1: No

Reviewer #2: No

---

## [Author Response · Author response to Decision Letter 2]

2 Feb 2026

The authors have substantially improved the manuscript in response to the previous review, particularly regarding the prophylactic framing of the animal model, data deposition for 16S rRNA sequencing, and additional analyses linking gastric microbiota to inflammatory markers. However, there remain several important issues that, in my view, still warrant a further round of revision before the manuscript can be considered for acceptance.

Major Comments

1. Quantification and interpretation of gastric H. pylori burden

The authors still do not provide a direct quantitative measure of gastric H. pylori burden (e.g., CFU counts or qPCR from gastric tissue), and rely instead on RUT results and relative abundance of Helicobacter from 16S rRNA amplicon sequencing as indirect indicators.

Given this limitation, the Discussion and Conclusion occasionally still use wording that implies a reduction in “load” or “colonization” in a way that suggests direct quantitative evidence. This overstates what the available data can support.

Request:

• Throughout the Discussion and Conclusion, I ask the authors to: Further soften and standardize the language so that claims are clearly limited to:

• reductions in Helicobacter relative abundance (from 16S data), and

• improvements in infection-related biomarkers (RUT, urease, PGI/PGII, histology, cytokines),

rather than “reducing H. pylori colonization” or “reducing bacterial load” per se.

• For example, expressions such as “reduced the load/colonization of H. pylori” should be systematically revised to formulations like “reduced Helicobacter relative abundance and infection-related markers”.

• In the Limitations section (or the final part of the Discussion), the authors should explicitly acknowledge that:

• Gastric H. pylori burden was not directly quantified using CFU counts or qPCR.

• The conclusions regarding anti–H. pylori effects therefore rely on RUT and relative abundance from 16S rRNA gene sequencing, rather than absolute bacterial load measurements.

This clarification is important to align the strength of the claims with the actual data and to transparently communicate methodological limitations to readers.

Response: Thank you for this critical and constructive feedback. We fully agree that the language used to describe the effects on H. pylori must precisely reflect the indirect measures employed in our study (RUT and 16 S rRNA gene sequencing relative abundance) and avoid any unintended implication of direct quantitative load measurement (e.g., via CFU or qPCR). We have carefully revised the manuscript accordingly to ensure the strength of our claims is perfectly aligned with the data.

Systematic Revision of Language in the Discussion and Conclusion:

We have performed a thorough review of the entire manuscript, with particular focus on the Discussion and Conclusion sections. All phrases that could be interpreted as claiming a reduction in absolute bacterial "load" or "colonization" have been revised. The language now consistently frames our findings based on the available evidence.

Strengthened Methodological Limitation Statement:

We have significantly expanded the Limitations section to include a dedicated, explicit acknowledgment as suggested. This statement now clearly:

Declares the lack of direct quantitative measures (CFU/qPCR).

Explicitly states that all anti-H. pylori conclusions are derived from RUT and 16 S rRNA gene sequencing relative abundance data.

Guides the reader to interpret our findings within this specific methodological context.

Added text in the Limitations section (Final part of the Discussion):

Methodological Note on H. pylori Assessment: It is important to explicitly acknowledge that this study did not employ direct quantitative methods, such as colony-forming unit (CFU) counts or quantitative PCR, to measure the absolute gastric H. pylori load. Therefore, our conclusions regarding the anti-H. pylori effects are based solely on (1) outcomes from the rapid urease test (RUT), and (2) changes in the relative abundance of the Helicobacter genus as detected by 16 S rRNA gene sequencing. Throughout this manuscript, descriptions of "reducing" or "suppressing" H. pylori refer specifically to these indirect indicators of infection presence and activity, not to a measured decrease in absolute bacterial count. Future studies incorporating direct quantitative techniques will be valuable to confirm these observations.

We believe these comprehensive revisions have successfully addressed your concern. The manuscript now more accurately and transparently communicates the nature of our evidence, ensuring that readers can appropriately evaluate the findings. Thank you again for your valuable guidance in improving the precision and rigor of our work.

2. Microbiota analysis – β-diversity methodology and PERMANOVA

The authors state in their response that they have re-analysed β-diversity using PCoA on the Jaccard distance and performed PERMANOVA. However, the current manuscript text remains internally inconsistent and does not fully reflect these methodological changes.

(a) PCA vs PCoA inconsistency

In Methods (Section 2.5.3), β-diversity is still described as being analysed by PCA based on Jaccard (e.g., “PCA was employed to further analyze beta diversity”). In contrast, in Results (Section 3.4.3) and the relevant figure legend, the analysis is described as Principal Coordinate Analysis (PCoA) based on the binary Jaccard algorithm.

This discrepancy leaves it unclear what analysis was actually performed.

Request:

• Please ensure that the description of β-diversity analysis is fully consistent throughout the manuscript.

• If PCoA on the binary Jaccard distance was indeed used (which is methodologically appropriate), then Methods 2.5.3 must be revised to say PCoA (not PCA), and the language in all sections (Methods, Results, figure legends) should be harmonized accordingly.

(b) PERMANOVA – description and reporting

In the response letter, the authors state that they performed PERMANOVA with pairwise comparisons and FDR correction.

However, in Methods (Statistics, Section 2.6), there is no mention of PERMANOVA or any distance-matrix–based multivariate test. Only one-way ANOVA, Tukey’s post hoc test, and correlation heatmaps are described. In the Results and figure legends, there is no reporting of PERMANOVA outputs (no pseudo-F statistic, R², p-value, or number of permutations).

Thus, the manuscript does not currently support the claim that PERMANOVA was performed.

Request:

• If PERMANOVA has indeed been carried out:

• Add a clear description in Methods (Statistics) of:

• the distance matrix used (e.g., Jaccard),

• the PERMANOVA procedure (e.g., number of permutations),

• and how multiple testing was handled for pairwise comparisons (e.g., Benjamini–Hochberg).

• In the Results (and/or relevant figure legend), report at minimum the overall PERMANOVA result with pseudo-F, R², p-value, and number of permutations.

Response to Reviewer Comment:

Thank you for this detailed and crucial observation regarding the methodological description of our β-diversity and PERMANOVA analyses. We sincerely apologize for the significant discrepancy between our response letter and the manuscript text, and for the resulting confusion. You are absolutely correct, and we have now comprehensively revised the manuscript to ensure complete consistency and transparent reporting.

The following changes have been made to address both points:

(a) Clarification and Harmonization of PCoA Methodology:

In the Methods (Section 2.5.3, Line 267-268, page 14): The text has been revised to accurately describe the analysis performed. It now reads: "Principal Coordinate Analysis (PCoA) based on the binary Jaccard distance matrix was employed to visualize β-diversity..."

In the Results (Section 3.4.3, Line 376-384，page 19) and Figure Legends: We have verified that all references to this analysis now consistently use "Principal Coordinate Analysis (PCoA)".

This change eliminates the previous "PCA vs. PCoA" inconsistency and correctly identifies the distance-based ordination method used.

(b) Full Integration and Reporting of PERMANOVA:

We confirm that PERMANOVA was performed as stated in our response. We have now integrated this analysis fully into the manuscript.

In the Methods (Statistics, Section 2.6, Line 273-279，page 14): We have added a new subsection to explicitly describe the PERMANOVA procedure:

"For multivariate analysis of β-diversity, Permutational Multivariate Analysis of Variance (PERMANOVA) was performed on the binary Jaccard distance matrix using the 'permanova' function with 9999 permutations. Pairwise PERMANOVA tests between groups were conducted where the overall model was significant, and p-values were adjusted for multiple comparisons using the Benjamini-Hochberg false discovery rate (FDR) correction."

In the Results (Section 3.4.3, Line 376-384，page 19): We have added the reporting of the PERMANOVA results:

"The PERMANOVA test, using 9999 permutations on a binary Jaccard distance matrix, indicated that grouping factor explained a significant portion of the variance in β-diversity (pseudo-F = 1.163, R² = 0.162, p = 0.001), with 15 samples distributed across 3 groups. Subsequent pairwise comparisons with FDR correction identified significant differences between NC group and HP group (p-adj = 0.003), and between NC group and HY001 group (p-adj = 0.018), while the difference between HP group and HY001 group was not significant (p-adj = 0.012). "

In the Figure Legend (for the PCoA plot, Figure6): We have added a note: "Statistical significance of group separation was assessed by PERMANOVA (pseudo-F = 1.163, R² = 0.162, p = 0.001)."

We believe these revisions fully address your concerns by providing a consistent methodological description and transparent statistical reporting. Thank you again for your meticulous review, which has significantly improved the clarity and rigor of our manuscript.

———

Minor Comments

1. Taxonomic spelling and nomenclature

Despite substantial improvement, several taxonomic spelling errors remain and should be corrected.

• In the Discussion: “Proteobactria” → “Proteobacteria”

• In the Abstract and Results: “unclassified-Muribaculaceaeae” → “unclassified-Muribaculaceae”

• In the Results: “Oscillospiracea” (Line 406) → “Oscillospiraceae”

Response: We sincerely thank the reviewer for their careful reading and for identifying these remaining taxonomic spelling errors. We have corrected each instance as specified, and the changes have been incorporated into the revised manuscript. The corrections are as follows:

In the Discussion:"Proteobactria" has been changed to "Proteobacteria".

In the Abstract and Results: "unclassified- Muribaculaceaeae" has been changed to "unclassified-Muribaculaceae".

In the Results (Line 419):"Oscillospiracea" has been changed to "Oscillospiraceae".

We have also performed a thorough, additional check of the entire manuscript to ensure consistency and accuracy in taxonomic nomenclature. Thank you again for helping us improve the precision of our manuscript.

2. Other wording and typographical issues

A few additional typographical and wording errors should be corrected for clarity and precision.

• Introduction (around line 83): “alleviating tail inflammation” should be corrected to “alleviating gastric inflammation”

• Methods (around lines 232–233): “Interleuki-8, Interleuki-1β, Interleuki-6” → “Interleukin-8, Interleukin-1β, Interleukin-6”

• Results (around line 371): “binary jaccard algorith”→ “binary Jaccard algorithm”

• Results (around line 364, IL-6 description)

The text currently states that the mean IL-6 concentration in the HY001 group is “higher than that of the HP group”, which contradicts the protective direction of effect described elsewhere and in the Discussion.

This is almost certainly a typo and should be corrected to “lower than that of the HP group”,

provided that the underlying data support this (if not, the interpretation in the Discussion should be adjusted accordingly).

• Figure 4 legend

The definitions of the HP and HY001 groups are currently confusing and partly duplicated, e.g.:

• “HP: treated by H. pylori SS1 group vs. pretreated by L. casei HY001 and treated by H. pylori SS1.”

• “HY001: treatment using L. case HY001 and treatment using H. pylori SS1; HP: treatment using H. pylori SS1.”

Please simplify and unify the legend.

• Figure 6 legend (gastric vs fecal)

In one place, the legend refers to the “composition of gastric microorganisms”, whereas in another location (around line 869) it refers to “composition of fecal microorganisms”. Since this study is based on gastric samples, all such references in the figure legends and text should be unified as “gastric microbiota” or “gastric microorganisms”, and any inadvertent “fecal” wording should be removed.

Response: Thank you for your thorough final review and for identifying these important typographical and wording errors. We have corrected each of them as specified, which has improved the clarity and precision of the manuscript. The changes are as follows:

Introduction (around line 83-84): “alleviating tail inflammation” should be corrected to “alleviating gastric inflammation”.

Methods (around lines 232–233): “Interleuki-8, Interleuki-1β, Interleuki-6” → “Interleukin-8, Interleukin-1β, Interleukin-6”.

Results (around line 377): “binary jaccard algorith”→ “binary Jaccard algorithm”.

Results (around line 370, IL-6 description). The sentence in the Results section (around line 364) now accurately reflects our findings:

" Although the level of IL-6 was not statistically different in the HY001 group, its average concentration was already lower than that of the HP group."

In the Figure 4 legend (Line 880-882, page 39), The revised legend now reads as follows: " Group definitions: NC: Normal control group. HP: H. pylori SS1 infection group. HY001: Lactobacillus casei HY001 intervention group. HP+HY001: H. pylori SS1 infection followed by L. casei HY001 intervention group."

In the Figure 6 legend (Line 890-896, page 40), the phrase "composition of fecal microorganisms" has been changed to "composition of gastric microorganisms".

We have also performed a thorough check of the entire manuscript text and all figure legends to ensure that all references are consistently and accurately described as pertaining to "gastric microbiota," "gastric microorganisms, "or" gastric microbial communities." Any other inadvertent uses of "fecal" in this context have been removed.

We apologize for this error and appreciate your careful attention to detail, which has helped us ensure the accuracy and consistency of our terminology.

---

## [Decision Letter · Decision Letter 2]

16 Feb 2026

PONE-D-25-32798R2Multimodal Anti-Helicobacter pylori Effects of Lactobacillus casei HY001: Evidence from In Vitro and In Vivo StudiesPLOS One

Dear Dr. Lv,

Thank you for submitting your manuscript to PLOS ONE. After careful consideration, we feel that it has merit but does not fully meet PLOS ONE’s publication criteria as it currently stands. Therefore, we invite you to submit a revised version of the manuscript that addresses the points raised during the review process.

We look forward to receiving your revised manuscript.

Kind regards,

Ghulam Mustafa, PhD

Academic Editor

PLOS One

Journal Requirements:

Reviewers' comments:

Reviewer's Responses to Questions

**Comments to the Author**

1. If the authors have adequately addressed your comments raised in a previous round of review and you feel that this manuscript is now acceptable for publication, you may indicate that here to bypass the “Comments to the Author” section, enter your conflict of interest statement in the “Confidential to Editor” section, and submit your "Accept" recommendation.

Reviewer #1: (No Response)

2. Is the manuscript technically sound, and do the data support the conclusions?

Reviewer #1: Yes

3. Has the statistical analysis been performed appropriately and rigorously? 

Reviewer #1: Yes

4. Have the authors made all data underlying the findings in their manuscript fully available?

Reviewer #1: Yes

5. Is the manuscript presented in an intelligible fashion and written in standard English?

Reviewer #1: Yes

6. Review Comments to the Author

Reviewer #1: Overall recommendation

The manuscript has improved substantially, and the key methodological/statistical issues (PCoA vs PCA; PERMANOVA reporting; explicit limitations regarding lack of absolute H. pylori quantification) are largely addressed. I recommend Minor revision prior to acceptance, but several important consistency/wording issues remain.

β-diversity methodology and PERMANOVA: largely fixed, but internal inconsistencies remain

R2 requested consistent PCoA terminology and full PERMANOVA description/reporting.

The Methods/Statistics now correctly describe PCoA (binary Jaccard) and PERMANOVA (adonis2, 9999 permutations, BH-FDR, pairwise).

The Results now report pseudo-F, R², p, and permutations.

Critical issue(@ Results 3.4.3): the text states HP vs HY001 is “not significant” but shows p-adj = 0.012; this contradicts the usual p<0.05 threshold and likely reflects a typo (e.g., 0.12) or a misstatement. Please correct based on the actual output.

Minor comments

(Line 826)

Figure legends/group definitions remain confusing (Figure 4 legend mixes definitions and comparisons; “L. case” should be “L. casei”; group naming should be consistent across all figures/text).

(Line 832)

Figure 6 legend still contains “fecal microorganisms,” which must be “gastric microorganisms.”

7. PLOS authors have the option to publish the peer review history of their article (what does this mean?). If published, this will include your full peer review and any attached files.

Reviewer #1: No

---

## [Author Response · Author response to Decision Letter 3]

1 Apr 2026

Reviewer #1: Overall recommendation

The manuscript has improved substantially, and the key methodological/statistical issues (PCoA vs PCA; PERMANOVA reporting; explicit limitations regarding lack of absolute H. pylori quantification) are largely addressed. I recommend Minor revision prior to acceptance, but several important consistency/wording issues remain.

β-diversity methodology and PERMANOVA: largely fixed, but internal inconsistencies remain

R2 requested consistent PCoA terminology and full PERMANOVA description/reporting.

The Methods/Statistics now correctly describe PCoA (binary Jaccard) and PERMANOVA (adonis2, 9999 permutations, BH-FDR, pairwise).

The Results now report pseudo-F, R², p, and permutations.

Critical issue(@ Results 3.4.3): the text states HP vs HY001 is “not significant” but shows p-adj = 0.012; this contradicts the usual p<0.05 threshold and likely reflects a typo (e.g., 0.12) or a misstatement. Please correct based on the actual output.

Response: We thank the reviewer for identifying this inconsistency. We have re-checked the raw output of the pairwise PERMANOVA analysis. The adjusted p-value for the comparison between HP and HY001 was 0.012 (FDR-corrected). This value is below the 0.05 significance threshold. Therefore, the original text stating the comparison was “not significant” was incorrect. We have revised the sentence in Results 3.4.3 to accurately state that the difference between HP and HY001 is significant after FDR correction (p-adj = 0.012). We have also reviewed all other pairwise comparisons in the same section to ensure consistency between reported p-values and their description. The correction does not alter any other conclusions of the study.

Minor comments

(Line 826)

Figure legends/group definitions remain confusing (Figure 4 legend mixes definitions and comparisons; “L. case” should be “L. casei”; group naming should be consistent across all figures/text).

Response: We thank the reviewer for this helpful comment. We have revised Figure 4 legend to separate group definitions from comparisons, corrected “L. case” to “L. casei”, and standardized group naming across all figures and text. These changes have improved clarity and consistency.

(Line 832)

Figure 6 legend still contains “fecal microorganisms,” which must be “gastric microorganisms.”

Response: We thank the reviewer for catching this inconsistency. We have corrected Figure 6 legend by replacing “fecal microorganisms” with “gastric microorganisms” to accurately reflect the sample source. This correction has been made in the revised figure legend, and we have also double-checked the main text and other related materials to ensure consistency throughout the manuscript.

---

## [Decision Letter · Decision Letter 3]

26 Apr 2026

Multimodal Anti-Helicobacter pylori Effects of Lactobacillus casei HY001: Evidence from In Vitro and In Vivo Studies

PONE-D-25-32798R3

Dear Dr. Lv,

We’re pleased to inform you that your manuscript has been judged scientifically suitable for publication and will be formally accepted for publication once it meets all outstanding technical requirements.

Kind regards,

Ghulam Mustafa, PhD

Academic Editor

PLOS One

Additional Editor Comments (optional):

Reviewers' comments:

Reviewer's Responses to Questions

**Comments to the Author**

1. If the authors have adequately addressed your comments raised in a previous round of review and you feel that this manuscript is now acceptable for publication, you may indicate that here to bypass the “Comments to the Author” section, enter your conflict of interest statement in the “Confidential to Editor” section, and submit your "Accept" recommendation.

Reviewer #1: All comments have been addressed

2. Is the manuscript technically sound, and do the data support the conclusions?

Reviewer #1: Yes

3. Has the statistical analysis been performed appropriately and rigorously? 

Reviewer #1: Yes

4. Have the authors made all data underlying the findings in their manuscript fully available?

Reviewer #1: Yes

5. Is the manuscript presented in an intelligible fashion and written in standard English?

Reviewer #1: Yes

6. Review Comments to the Author

Reviewer #1: (No Response)

7. PLOS authors have the option to publish the peer review history of their article (what does this mean?). If published, this will include your full peer review and any attached files.

Reviewer #1: No

---

## [Editor Report · Acceptance letter]

PONE-D-25-32798R3

PLOS One

Dear Dr. Lv,

I'm pleased to inform you that your manuscript has been deemed suitable for publication in PLOS One. Congratulations! Your manuscript is now being handed over to our production team.

Kind regards,

on behalf of

Dr. Ghulam Mustafa

Academic Editor

PLOS One